# CROSSING THE SEPARATION POINT: STABILIZING DECISION-FOCUSED LEARNING WITH VARIATIONAL FREE-ENERGY

## ABSTRACT

Decision-focused learning (DFL) integrates predictive modeling with downstream optimization by training the prediction model to end-to-end minimize decision regret. However, the mapping from predicted parameters to decisions can be discontinuous, making gradients unstable and optimization unstable, particularly in the early stages of training. Existing approaches typically rely on heuristic warm-up phases through pretraining for stabilization, but the lack of theoretical understanding limits their effectiveness and generalizability. In this work, we propose a principled framework based on variational free energy (VFE) to analyze and address these challenges. We characterize the existence of critical separation points in the training dynamics, which mark a sharp transition in the alignment between prediction accuracy and decision quality. Building on this analysis, we develop Adaptive Recursive Annealing (ARA), a training strategy that adaptively regulates the learning dynamics without requiring warm-up. ARA improves training stability and decision performance by aligning upstream and downstream objectives throughout the learning process. Experiments on multiple benchmark problems demonstrate that our method consistently improves convergence behavior and downstream decision quality, offering a robust and theoretically grounded alternative to existing heuristics.

## 1 INTRODUCTION

Deep learning models are primarily designed for predictive tasks, where the objective is to minimize a loss function that quantifies the difference between predicted and observed values. This predictive-focused learning paradigm has proven effective in domains where prediction quality directly governs downstream performance. However, in many applications such as resource allocation, scheduling, or planning, the final goal is to make decisions based on these predictions. In such settings, high predictive accuracy does not always translate into high decision quality.(Elmachtoub & Grigas, 2022; Mandi et al., 2020; Hu et al., 2023).

To bridge this gap, *decision-focused learning* (DFL) was introduced (Wilder et al., 2019; Pogančić et al., 2020). DFL integrates the prediction model with a downstream optimization problem and updates model parameters using the decision loss. Rather than optimizing prediction accuracy alone, DFL aligns training with the quality of the decisions induced. This paradigm has gained traction in energy management (Geng et al., 2023), logistics (Qi et al., 2023), public health (Wang et al., 2023), and finance (Wang et al., 2020). Despite its promise, DFL poses a significant technical challenge: in most optimization problems, the decision loss is piecewise constant or non-differentiable, making gradient-based learning unreliable (Mulamba et al., 2021; Ferber et al., 2020).

The non-smoothness properties of the loss create difficulties for gradient-based learning. Gradients from decision loss can be sparse, unstable, or misleading in early training when predictions are far from optimal. As a result, many methods adopt a two-phase heuristic: first train the model using a prediction loss such as MSE, then fine-tune using the decision loss (Mandi et al., 2022; Geng et al., 2023; Ma et al., 2024). While this warm-up phase can improve convergence, it lacks theoretical grounding and introduces sensitive hyperparameters (Shah et al., 2022a). More fundamentally, it reflects an open question in the field: *the learning dynamics of DFL remain poorly understood.*

Understanding how prediction and decision objectives interact during training is increasingly recognized as a critical problem in deep learning (Kolter & Manek, 2019; Zhao et al., 2024; Jiang et al., 2024). In this work, we identify a specific failure mode that arises in DFL, which we call *separation point instability*. This describes scenarios where slight improvements in prediction accuracy lead to worse decisions, due to crossing discontinuities in the decision map. We show that this phenomenon occurs at sharp transitions in the argmin mapping of the optimization layer.

To analyze this effect, we draw inspiration from the principle of *variational free energy*—a theoretical tool grounded in statistical physics and Bayesian inference (Friston et al., 2023; Sidky & Whitmer, 2018; Shen et al., 2025). Free energy models have been used to understand dynamics in neural systems and physical processes, and we adapt this lens to study decision-focused learning. Our framework reveals that separation points correspond to high-energy boundaries in the loss landscape where training becomes unstable. This leads to the following research questions:

- **RQ1:** Can we formally define separation points and characterize their role in decision-focused instability?

- **RQ2:** Can a variational free-energy framework offer theoretical insights into training dynamics near such points?

- **RQ3:** Can we design adaptive strategies that improve stability without relying on warm-up heuristics?

To address these questions, we make the following contributions:

1. We provide a formal characterization of separation points based on an energy-theoretic view of the decision loss landscape, and prove a sufficient condition for their emergence.

2. We introduce a novel framework that aligns the upstream predictive and downstream decision objectives through a variational free-energy formulation.

3. We develop an adaptive training method, Adaptive Recursive Annealing (ARA), which dynamically adjusts optimization schedules based on signals derived from the variational energy landscape. Unlike heuristic warm-up, ARA is theoretically grounded and aligns closely with the conditions that trigger separation point instability.

## 2 CHALLENGE: LEARNING INSTABILITY AND THE ROLE OF WARM-UP

In decision-focused learning (DFL), model training is guided not by prediction accuracy alone, but by the quality of the downstream decisions induced by the predictions. The standard setup involves learning a cost estimator $\hat{c} \in \mathbb{R}^d$ for an unknown true cost $c \in \mathbb{R}^d$, which is then used to solve the linear optimization problem:

$$w^*(\hat{c}) = \arg \min_{w \in \mathcal{W}} \hat{c}^\top w, \tag{1}$$

where $\mathcal{W}$ is a finite or polyhedral feasible region, and deterministic tie-breaking ensures $w^*$ is unique.

In this context, the performance of the learning model is commonly evaluated from two perspectives. One is the **Prediction loss**, we use mean squared error (MSE) as a representative surrogate, defined as $\tilde{L}_{\text{pred}}(\hat{c}, c) = \|\hat{c} - c\|^2$. Note that other loss forms, such as MAE or KL divergence, may also be appropriate depending on the application. The other one is the **Decision loss** $\tilde{L}_{\text{dec}}(\hat{c}, c) = c^\top w^*(\hat{c}) - c^\top w^*(c)$, which directly measures the suboptimality of the predicted decision.

While the prediction loss is smooth and differentiable, the decision-focused loss is not. The map $\hat{c} \mapsto w^*(\hat{c})$ is piecewise-constant and may change discontinuously when $\hat{c}$ crosses decision boundaries. As a result, small improvements in prediction accuracy can lead to sharp deterioration in decision quality—a phenomenon we refer to as the *separation point instability*. When a separation point occurs, it indicates an inconsistency in learning dynamics, where optimizing one loss harms the other. Now, we define the separation point as follows.

**Definition 2.1** (Separation Point). Let $v \in \mathbb{R}^d$ be a unit direction and let all gradients be taken with respect to the prediction $\hat{c}$. A prediction $\hat{c}$ is said to lie at a **separation point** along direction $v$ if an

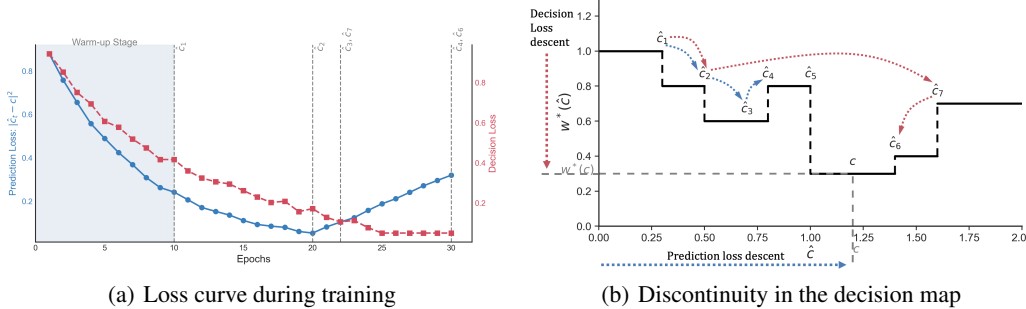

(a) Loss curve during training          (b) Discontinuity in the decision map

Figure 1: (a) During early training, prediction and decision losses decrease together, which is often referred to as the "warm-up" stage. However, once $\hat{c}$ approaches a decision boundary (e.g., $\hat{c}_1$), the two losses begin to diverge. (b) A piecewise-constant decision map: small movements in $\hat{c}$ (from $\hat{c}_1$ to $\hat{c}_2$, etc.) can cause sharp changes in $w^*(\hat{c})$, resulting in decision regret spikes even when prediction improves.

infinitesimal update of the form $\hat{c} + \epsilon v$ simultaneously decreases the prediction loss but increases the decision loss. Formally,

$$\left\langle \nabla_{\hat{c}} \tilde{L}_{\text{pred}}(\hat{c}, c), v \right\rangle \cdot \left\langle \nabla_{\hat{c}} \tilde{L}_{\text{dec}}(\hat{c}, c), v \right\rangle < 0.$$

Here $\nabla_{\hat{c}} \tilde{L}_{\text{pred}}$ and $\nabla_{\hat{c}} \tilde{L}_{\text{dec}}$ denote the gradients of the prediction and decision losses with respect to the predicted cost vector $\hat{c}$.

Due to the instability near such separation points, most DFL algorithms employ a *warm-up phase*, where the model is pretrained using a smooth prediction loss before switching to the decision-focused loss. This approach improves training stability and helps avoid convergence failure in the early stages. However, it raises several unresolved questions. **(1). When and why is warm-up necessary? (2). How many epochs of pretraining are sufficient or optimal? (3). Can the training process be stabilized without relying on this heuristic?**

These questions point to a more fundamental gap: the absence of a rigorous understanding of how prediction errors interact with decision boundaries during training. We find that learning trajectories frequently pass through regions where the decision map changes discretely, leading to sharp increases in the decision loss even as predictive accuracy improves. This motivates the **key challenge** addressed in this work:

> *Decision-focused learning suffers from instability due to discontinuities in the decision map. These instabilities undermine the effectiveness of gradient-based optimization. Current methods rely on warm-up heuristics to bypass early-stage instability, but lack a theoretical foundation to explain, predict, or eliminate such behavior. A principled analysis is required to uncover the learning dynamics near separation points and guide the design of stable, adaptive training algorithms.*

In the following section, we introduce a formal characterization of separation points based on the geometry of the optimization landscape. Building on this insight, we develop an energy-based framework that provides both theoretical clarity and algorithmic stability.

## 3 SEPARATION POINTS IN DECISION-FOCUSED LEARNING

### 3.1 PERTURBATIONS NEAR A DECISION BOUNDARY

Consider a prediction $\hat{c}_0$ at a given training step. To test stability, we perturb it in a direction $v \in \mathbb{R}^d$,

$$\hat{c} = \hat{c}_0 + \epsilon v, \qquad \epsilon > 0.$$

If the two losses remain aligned under this perturbation, training can proceed safely. If they move in opposite directions, the trajectory has entered an unstable regime. This behaviour typically occurs

when $\hat{c}_0$ lies on, or very close to, a decision boundary where two adjacent minimizers exchange optimality. At such a boundary point $x \in \mathbb{R}^d$ there are two different minimizers $w_a, w_b \in \mathcal{W}$ with

$$x^\top w_a = x^\top w_b = \min_{w \in \mathcal{W}} x^\top w.$$

The difference $d = w_a - w_b$ captures how sensitive the decision is along the boundary. Its projection on the probing direction is $\Delta = v^\top d$. When $\Delta \neq 0$, even an infinitesimal step along $v$ flips the optimal decision from $w_a$ to $w_b$ or vice versa.

To see the effect on the optimization landscape, define the value function $\Phi(x) = x^\top w^*(x)$, where $w^*(x) \in \arg\min_{w \in \mathcal{W}} x^\top w$ denotes an optimal solution at cost $x$. Along the direction $v$, the one-sided directional derivative of $\Phi$ at the boundary point $x$ is

$$\partial_v^+ \Phi(x) = \begin{cases} v^\top w_b, & \Delta > 0, \\ v^\top w_a, & \Delta < 0. \end{cases}$$

Therefore, the gradient of $\Phi$ with respect to $x$ experiences a kink when the trajectory crosses the boundary: an arbitrarily small change in $\hat{c}$ can cause a jump from $v^\top w_a$ to $v^\top w_b$. This discontinuity in the directional derivative explains why decision-related losses can change abruptly even when the prediction error itself continues to decrease smoothly.

### 3.2 THE SCALED SEPARATION CRITERION

Prediction and decision losses typically evolve on different scales, so their raw magnitudes cannot be compared directly. We therefore maintain running normalization factors $\sigma_{\text{pred}}, \sigma_{\text{dec}} > 0$, defined as exponential moving estimates of the standard deviation of $L_{\text{pred}}$ and $L_{\text{dec}}$ over training batches. Based on these quantities, we construct normalized losses

$$L_{\text{pred}}(\hat{c}, c) := \frac{\tilde{L}_{\text{pred}}(\hat{c}, c)}{\sigma_{\text{pred}}}, \qquad L_{\text{dec}}(\hat{c}, c) := \frac{\tilde{L}_{\text{dec}}(\hat{c}, c)}{\sigma_{\text{dec}}},$$

and define a joint Lyapunov function

$$V_\lambda(\hat{c}) = \lambda\, L_{\text{pred}}(\hat{c}, c) + L_{\text{dec}}(\hat{c}, c), \qquad \lambda > 0.$$

This rescaling affects only the relative scale of the two losses. The scaled separation criterion in Theorem 3.1 is stated in terms of the unnormalized losses $L_{\text{pred}}$ and $L_{\text{dec}}$ and therefore does not depend on the particular choice of $\sigma_{\text{pred}}$ and $\sigma_{\text{dec}}$.

Consider now the situation where the true cost vector $c$ lies exactly on a decision boundary. Let $w_a, w_b \in \mathcal{W}$ be two distinct minimizers with $c^\top w_a = c^\top w_b$, and denote their difference by $d = w_a - w_b$. For a perturbation direction $v$, the signed displacement across the boundary is $\Delta = v^\top d$. After scaling, the effective gap becomes $\widetilde{\Delta} = \frac{\sigma_{\text{pred}}}{\sigma_{\text{dec}}} \Delta$. These ingredients lead to the following scaled separation criterion (proof in Appendix C.1):

**Theorem 3.1** (Scaled Separation Point Criterion). *Assume the true cost lies on a boundary with two distinct minimizers $w_a, w_b \in \mathcal{W}$ and let $d = w_a - w_b$. For a current prediction $\hat{c}_0$ and perturbation direction $v$, define $\Delta = v^\top d$ and $B = v^\top(\hat{c}_0 - c) - \frac{1}{2}\frac{\sigma_{\text{pred}}}{\sigma_{\text{dec}}}\Delta$. Then:*

*(a) The directional derivative of the prediction loss is $\partial_v L_{\text{pred}}(\hat{c}_0, c) = \frac{2}{\sigma_{\text{pred}}} v^\top(\hat{c}_0 - c)$.*

*(b) The directional derivative of the decision loss is $\partial_v^+ L_{\text{dec}}(\hat{c}_0, c) = -\frac{\Delta}{\sigma_{\text{dec}}} \operatorname{sgn}(B)$.*

*(c) There exists a weight $\lambda^\star = \frac{\sigma_{\text{pred}}}{2\sigma_{\text{dec}}}\left(1 + \frac{\widetilde{\Delta}}{2B}\right)\operatorname{sgn}(B)$ such that the Lyapunov function satisfies*

$$\partial_v V_{\lambda^\star}(\hat{c}_0) = 2B\,\operatorname{sgn}(B).$$

*(d) The hyperplane $\mathcal{H}_{\text{scaled}} = \{\hat{c} : v^\top(\hat{c} - c) = \frac{1}{2}\frac{\sigma_{\text{pred}}}{\sigma_{\text{dec}}}\Delta\}$ separates the training space: if $B > 0$ both losses and $V_{\lambda^\star}$ decrease along $+v$, whereas if $B < 0$ the two losses move in opposite directions and $V_{\lambda^\star}$ increases.*

*Remark* 3.2 (Scaled separation criterion in the loss plane). In the $(L_{\text{pred}}, L_{\text{dec}})$ plane, the normalized losses define a rotated coordinate system in which each contour $V_\lambda(\hat{c}) = \lambda L_{\text{pred}}(\hat{c}) + L_{\text{dec}}(\hat{c})$ corresponds to a straight line with slope $dL_{\text{dec}}/dL_{\text{pred}} = -\lambda$. In this plane, Theorem 3.1 states that there exists a slope $-\lambda^\star$ such that $V_\lambda(\hat{c}t)$ decreases whenever $L_{\text{pred}}(\hat{c}t) > L_{\text{dec}}(\hat{c}t)$, and the separation point is the location where $\lambda^\star L\text{pred} = L_{\text{dec}}$. In practice, stochastic optimization frequently drives $\hat{c}t$ along a narrow corridor in this plane until it hits this intersection, which is precisely where we observe the separation phenomena illustrated in Fig. 1. In Section 4, we construct the RVFE objective so that its level sets play the role of such contours while smoothing the decision map, ensuring that training continues to follow a decreasing $V_\lambda$-like functional even when the hard decision loss is locally flat.

This theorem clarifies the role of the Lyapunov function: by selecting an appropriate scaling $\lambda^\star$, it provides a descent certificate on the stable side of the separation hyperplane. Instability, therefore, is not incidental but follows inevitably from the geometry of decision boundaries once prediction and decision losses are placed on a common scale.

In the remainder of the paper, we treat $V_{\lambda^\star}$ as the ideal target that captures the right trade-off between prediction and decision quality, but we never optimize $V_{\lambda^\star}$ directly. Instead, we construct a smooth surrogate whose first term coincides with $L_{\text{pred}}$, and whose second term is a softened version of $L\text{dec}$. The relative weight in $V_{\lambda^\star}$ is controlled to mirror the scale $\lambda^\star$ identified in Theorem 3.1.

## 4 STABILIZING CONSISTENCY WITH RECURSIVE VARIATIONAL FREE ENERGY

The separation theorem tells us why training in decision-focused learning can suddenly go off track: once the iterate approaches a decision boundary, prediction and decision losses pull in opposite directions. What we need is a mechanism that keeps the trajectory away from these unreliable zones while still letting the model learn from them. Our answer is the *Recursive Variational Free Energy (RVFE)* framework, which reshapes the training objective smoother, and adaptively balanced.

### 4.1 VARIATIONAL RELAXATION THROUGH FREE ENERGY

The instability arises from the discontinuous map $\hat{c} \mapsto w^*(\hat{c})$: a tiny perturbation in $\hat{c}$ can flip the minimizer and cause a jump in the decision loss. To smooth this landscape, we replace the hard $\arg\min$ rule with a Gibbs distribution that assigns each decision $w \in \mathcal{W}$ a probability depending on its cost under $\hat{c}$:

$$P_\tau(w \mid \hat{c}) = \frac{\exp(-\tau \hat{c}^\top w)}{\sum_{u \in \mathcal{W}} \exp(-\tau \hat{c}^\top u)}, \qquad \tau > 0.$$

Here $\tau$ plays the role of a temperature: at high $\tau$ the distribution spreads across many decisions, while at low $\tau$ it concentrates on the minimizer. Using $P_\tau$, the decision loss becomes a smooth expectation,

$$L_{\text{dec}}^{\text{soft}}(\hat{c}, c) = \mathbb{E}_{w \sim P_\tau}[c^\top w] - c^\top w^*(c),$$

which eliminates discontinuous jumps and yields gradients defined everywhere. To further balance prediction and decision terms, we introduce a scaled version of this Gibbs rule, denoted $P_\beta(w \mid \hat{c})$, where the scaling parameter $\beta > 0$ controls the relative emphasis on prediction versus decision quality.

**Proposition 4.1.** *Let $\beta > 0$. Define the scaled Gibbs distribution*

$$P_\beta(w \mid \hat{c}) = \frac{\exp(-\hat{c}^\top w/(\beta \sigma_{\text{dec}}))}{\sum_{u \in \mathcal{W}} \exp(-\hat{c}^\top u/(\beta \sigma_{\text{dec}}))}.$$

*Then the free-energy objective admits the representation*

$$\mathcal{F}(\hat{c}; \beta) = \frac{\|\hat{c} - c\|^2}{\sigma_{\text{pred}}} + \frac{1}{\beta \sigma_{\text{dec}}} \Big[ c^\top \bar{w}_\beta(\hat{c}) - c^\top w^*(c) \Big],$$

*where $\bar{w}_\beta(\hat{c}) = \mathbb{E}_{P_\beta(\cdot|\hat{c})}[w]$ is the Gibbs-weighted average decision.*

This decomposition makes explicit how the free energy realizes the Lyapunov objective introduced in Section 3.2. Define the scaled prediction and (smoothed) decision losses

$$L_{\text{pred}}(\hat{c}, c) := \frac{\|\hat{c} - c\|^2}{\sigma_{\text{pred}}}, \qquad \tilde{L}_{\text{dec}}(\hat{c}, c) := \frac{1}{\sigma_{\text{dec}}}\Big(c^\top \bar{w}_\beta(\hat{c}) - c^\top w^\star(c)\Big),$$

where $\bar{w}_\beta(\hat{c}) = \mathbb{E}_{P_\beta(\cdot|\hat{c})}[w]$ is the Gibbs-weighted average decision. With this notation, Proposition 4.1 can be rewritten as

$$F(\hat{c}; \beta) = L_{\text{pred}}(\hat{c}, c) + \frac{1}{\beta} \tilde{L}_{\text{dec}}(\hat{c}, c) = \frac{1}{\beta}\Big(\beta\, L_{\text{pred}}(\hat{c}, c) + \tilde{L}_{\text{dec}}(\hat{c}, c)\Big).$$

Up to the positive scaling factor $1/\beta$, the bracket coincides with the Lyapunov combination

$$V_\lambda(\hat{c}; c) = \lambda\, L_{\text{pred}}(\hat{c}, c) + \tilde{L}_{\text{dec}}(\hat{c}, c)$$

for the specific choice $\lambda = \beta$. Consequently, the inverse temperature $\beta$ controls the same trade-off between prediction and decision losses as the Lyapunov weight $\lambda$: increasing $\beta$ emphasizes prediction accuracy, while decreasing $\beta$ places more relative weight on decision regret. In this sense, the free energy $F(\hat{c}; \beta)$ is a smooth realization of the Lyapunov trade-off prescribed by Theorem 3.1, with $\beta$ acting as the effective weight on the prediction term in $V_\lambda$.

## 4.2 MARGINS THAT PUSH AWAY FROM INDECISION

Smoothing the decision map reduces jumps, but near a boundary the Gibbs distribution still spreads probability mass over multiple candidates. In this region, the predictor is indecisive, and the softened objective may drift toward poor solutions. To discourage this behavior, we augment the free energy with a margin penalty that grows with uncertainty:

$$\delta_{\mu,\tau}(w \mid \hat{c}) = \mu\, \|w - \bar{w}\|_2, \qquad \bar{w} = \mathbb{E}_{P_\tau}[w].$$

Then we obtain a *margin–augmented free energy* by modifying the decision term of the free energy:

$$\mathcal{F}_{\tau,\mu}(\hat{c}; \beta) = \frac{\|\hat{c} - c\|^2}{\sigma_{\text{pred}}} + \frac{1}{\beta\sigma_{\text{dec}}}\Big(\mathbb{E}_{P_\tau}[c^\top w + \delta_{\mu,\tau}(w \mid \hat{c})] - c^\top w^*(c)\Big).$$

The margin induces a geometric "repulsion" from boundaries. Let $w_a, w_b \in \mathcal{W}$ be two competing minimizers at a boundary and set $d := w_a - w_b$. With $\langle \cdot, \cdot \rangle$ denoting the standard Euclidean inner product in $\mathbb{R}^d$. Then the gradient of $\mathcal{F}_{\tau,\mu}$ along $d$ satisfies the following.

In summary, RVFE replaces the discontinuous decision rule $w^\star(\hat{c})$ by the Gibbs-smoothed average $\bar{w}_\beta(\hat{c})$ and augments the resulting free energy with a variance-based margin. The three terms in $F_{\tau,\mu}(\hat{c}; \beta)$ play complementary roles: the MSE term pulls $\hat{c}$ toward the true cost $c$, the Gibbs-based regret term provides a smooth version of the Lyapunov decision loss and implements the trade-off weight $\lambda$ through $\beta$, and the margin term adds a repulsive force whenever the predictor is indecisive and probability mass is spread across multiple candidates. Together, they realize a smooth Lyapunov objective whose level sets avoid the unstable regions identified by the scaled separation criterion.

In all experiments, we use the margin–augmented free energy $F_{\tau,\mu}(\hat{c}; \beta)$ as the training loss of the predictive network and minimize its Monte Carlo estimate with respect to the network parameters by stochastic gradient descent, while the annealing parameters $(\beta_t, \tau_t)$ are updated according to the rule in Sec. 3.2 so that the descent direction adapts to the local separation geometry.

**Proposition 4.2.** *If $P_\tau(\cdot \mid \hat{c})$ assigns positive mass to both $w_a$ and $w_b$, then*

$$\langle \nabla_{\hat{c}} \mathcal{F}_{\tau,\mu}(\hat{c}; \beta), d \rangle = -\frac{\tau\mu}{\beta\sigma_{\text{dec}}} \text{Var}_{P_\tau}(\langle w, d \rangle) < 0.$$

Hence, whenever the predictor is uncertain, the free-energy gradient acquires a strictly negative component that repels $\hat{c}$ away from the decision boundary. The more uncertain the predictor, the stronger this repulsive force.

*Remark* 4.3 (Practical computation). For large decision sets $\mathcal{W}$, evaluating expectations under $P_\tau$ exactly is infeasible. We therefore approximate $P_\tau$ by the distribution $Q_\tau$ induced by perturb-and-MAP sampling, which reuses downstream solvers and naturally fits combinatorial optimization. In

practice, we draw $K$ i.i.d. Gumbel perturbations, compute the corresponding MAP solutions, and form empirical averages of these samples to approximate the expectations that appear in the gradient estimator. To quantify the discrepancy between $Q_\tau$ and $P_\tau$ we use the total variation distance $\text{TV}(Q_\tau, P_\tau)$. In all experiments, we use $K = 32$ samples per forward pass, which was sufficient to obtain stable gradients. The resulting runtime tables in the experiments show that RVFE remains competitive with existing decision-focused methods when using these approximate Gibbs samples.

The approximation introduces a distributional gap between $Q_\tau$ and $P_\tau$. The following result controls its effect on gradients.

**Proposition 4.4.** *Let $Q_\tau$ be the distribution induced by perturb-and-MAP. Assume decisions are coordinatewise bounded, $|w_j| \leq 1$ for all $w \in \mathcal{W}$ and $j$, and $\|f\|_\infty \leq M$. Then*

$$\left\| \mathbb{E}_{Q_\tau}[\, wf(w)\,] - \mathbb{E}_{P_\tau}[\, wf(w)\,] \right\|_2 \leq 2M \, \text{TV}(Q_\tau, P_\tau).$$

*Consequently, the gradient bias incurred by replacing $P_\tau$ with $Q_\tau$ is bounded by*

$$\left\| \frac{\tau}{\beta \sigma_{\text{dec}}} \left( \mathbb{E}_{Q_\tau}[\, wf(w)\,] - \mathbb{E}_{P_\tau}[\, wf(w)\,] \right) \right\|_2 \leq \frac{2\tau M}{\beta \sigma_{\text{dec}}} \, \text{TV}(Q_\tau, P_\tau),$$

*which vanishes as $Q_\tau \to P_\tau$. With $K$ samples from $Q_\tau$, the total error equals this bias plus an $\mathcal{O}(1/K)$ variance term.*

### 4.3 ANNEALING GUIDED BY SEPARATION POINTS

The last ingredient is to adapt the weights dynamically. Recall from Theorem 3.1 that the signed gap

$$B_t = v^\top (\hat{c}_t - c) - \tfrac{1}{2} \frac{\sigma_{\text{pred}}}{\sigma_{\text{dec}}} \Delta$$

tells us whether we are in the stable half-space ($B_t \geq 0$) or the trade-off half-space ($B_t < 0$). We use $B_t$ to update the balance between prediction and decision terms:

$$\beta_{t+1} = \begin{cases} \gamma_\downarrow \beta_t, & B_t \geq 0, \\ \gamma_\uparrow \beta_t, & B_t < 0, \end{cases} \qquad \tau_{t+1} = \alpha \tau_t.$$

Here $\gamma_\downarrow < 1$ gradually strengthens the decision term in stable zones, while $\gamma_\uparrow > 1$ reduces its weight when training turns into unstable zones. The smoothing temperature $\tau$ grows slowly to sharpen the surrogate.

**Theorem 4.5** (Monotone descent in the stable region). *Let $\{\hat{c}_t\}$ follow gradient descent on*

$$\mathcal{F}_{\text{soft}}(\hat{c}; \beta_t) = \|\hat{c} - c\|^2 / \sigma_{\text{pred}} + L_{\text{dec}}^{\text{soft}}(\hat{c}, c; \beta_t),$$

*with step size $\eta_t \leq 1/L_t$ and the above annealing rule. If $B_t \geq 0$ then*

$$\mathcal{F}_{\text{soft}}(\hat{c}_{t+1}; \beta_{t+1}) \leq \mathcal{F}_{\text{soft}}(\hat{c}_t; \beta_t) - \frac{\eta_t}{2} \|\nabla \mathcal{F}_{\text{soft}}(\hat{c}_t; \beta_t)\|^2.$$

This theorem formalizes the intuition: as long as the model remains in the stable region, the free energy decreases monotonically. Together, Proposition 4.1, Proposition 4.2, and Theorem 4.5 explain how RVFE smooths the landscape, pushes the predictor away from boundaries, and guarantees well-behaved descent when training stays on the right side of the separation point.

## 5 EXPERIMENTS

We evaluate RVFE on five predictive combinatorial optimization tasks, aimimg to answer:

- **Q1:** Does RVFE consistently improve decision quality over existing methods?
- **Q2:** Can RVFE bypass the need for MSE-based pretraining?
- **Q3:** How sensitive are other methods to pretraining, and how robust is RVFE?

## 5.1 EXPERIMENTAL SETTINGS

We evaluate our approach on six widely used decision-focused benchmarks: Energy Scheduling (Energy) (Mandi et al., 2022; Mandi & Guns, 2020; Mandi et al., 2020; Mulamba et al., 2021), Bipartite Matching (Bimatching) (Ferber et al., 2020), Knapsack (Tang & Khalil, 2022), Budget Allocation (BudgetAlloc) (Wilder et al., 2019), Portfolio Optimization (Wang et al., 2020; Shah et al., 2022b), and Cubic Top-$k$ (Cubic) (Geng et al., 2023). Each problem is formulated as a structured prediction task in which a feature vector is mapped to a cost estimate $\hat{c}$, which in turn drives a downstream optimization.

These benchmarks were chosen deliberately to cover complementary aspects of decision-focused learning. Energy, BudgetAlloc, and Portfolio capture real-world resource allocation tasks with continuous or softly constrained feasible regions, where smooth regret landscapes are expected. Bimatching and Knapsack test combinatorial settings with sharp feasibility constraints, probing the stability of optimization under discrete jumps. Finally, Cubic serves as a stress test: it deliberately violates the smoothness assumptions by combining nonlinear prediction targets with sorting-based, discontinuous decision rules.

We compare the proposed RVFE against several classes of methods: *Surrogate-based*, including SPO+ (Elmachtoub & Grigas, 2022), DFL (Wilder et al., 2019), BlackBox(Pogančić et al., 2020), NCE(Mulamba et al., 2021). *Ranking-based*(Mandi et al., 2022), including PointLTR, PairLTR, ListLTR. All methods use the same neural architecture (2-layer ReLU network, hidden size 32) and are trained using Adam optimizer with early stopping and batch size 64. Each baseline is evaluated with 0–20 epochs of MSE pretraining, denoted as Pre_x (e.g., Pre_10 for 10 pretraining epochs). RVFE does not use pretraining and instead applies adaptive annealing.

Unless stated otherwise, Table 1 reports the relative decision regret on the test set. Given the true cost vectors $\{c_i\}$ and the predicted-cost decisions $w^*(\hat{c}_i)$, we evaluate the true objective on both the predicted decision and the oracle decision $w^*(c_i)$. Lower values indicate closer alignment with the oracle solutions. The evaluation Regret metric is

$$\text{Regret\%} = 100 \times \frac{\sum_i \big| c_i^\top w^*(\hat{c}_i) - c_i^\top w^*(c_i) \big|}{\sum_i \big| c_i^\top w^*(c_i) \big|}.$$

## 5.2 MAIN RESULTS: IMPACT OF PRETRAINING AND ANNEALING

Table 1 reports the final regret of six methods on six benchmark tasks. For each baseline, we vary the number of pretraining steps from 0 to 20. For RVFE, we evaluate both the default version (with annealing) and a variant without it. The results reveal several patterns. First, pretraining is beneficial for enhancing the overall performance of most baselines across various benchmarks. For example, the regret of DFL, BB, and NCE decreases as the number of warm-up epochs increases. As for RVFE, it performs well across most benchmarks without pretraining. For instance, in the Portfolio task, RVFE outperforms all baselines, including those trained with 15-step pretraining. The same trend holds in Energy, BudgetAlloc, and Portfolio.

**Answer to Q1:** RVFE achieves lower regret across most benchmark tasks, outperforming baselines, even when some of them are pretrained. This confirms its advantage in decision quality.

**Answer to Q2:** RVFE utilizes the ARA mechanism to replace the heuristic warm-up approach. Results from extensive experiments show that warm-up is not necessary for stable convergence.

## 5.3 ANNEALING STABILITY: SENSITIVITY TO HYPERPARAMETERS

We further investigate how sensitive RVFE is to its internal parameters. In particular, we vary the annealing rates $\gamma_\downarrow$ and $\gamma_\uparrow$ that control how the variational coefficients $\beta$ and $\tau$ evolve over time. Figure 2 shows the regret values as a function of these two hyperparameters for each task. The results are consistent. Across all six tasks, RVFE maintains low regret in a broad region of the parameter space. Most tasks reach optimal performance near $(\gamma_\downarrow = 0.5, \gamma_\uparrow = 2.0)$, but neighboring values still yield competitive outcomes. This suggests that the method is not overly sensitive to fine-tuning and can be deployed with a default configuration.

Table 1: Final regret (%, lower is better) for all methods across six tasks.

| Method | Config | Energy | Bimatch | Knapsack | BudgetAlloc | Portfolio | Cubic |
|---|---|---|---|---|---|---|---|
| MSE | Pre_0 | 6.03±0.43 | 90.79±0.66 | **6.56**±0.76 | 19.83±2.00 | 0.28±0.05 | 0.22±0.02 |
| | Pre_5 | 5.38±0.32 | 90.66±0.56 | 6.91±0.67 | 19.13±1.64 | 0.24±0.04 | **0.20**±0.01 |
| | Pre_10 | 5.18±0.29 | **90.77**±0.50 | 7.31±0.64 | 19.03±1.46 | 0.23±0.03 | **0.20**±0.01 |
| | Pre_20 | **4.65**±0.23 | 90.54±0.45 | 7.71±0.61 | **18.71**±1.29 | **0.23**±0.02 | 0.20±0.01 |
| SPO+ | Pre_0 | 1.57±0.12 | **90.95**±0.71 | **6.14**±0.78 | 5.49±0.61 | 0.26±0.05 | 160.60±14.99 |
| | Pre_5 | 1.53±0.10 | 92.23±0.60 | 6.51±0.69 | 5.48±0.50 | 0.23±0.04 | 148.43±11.74 |
| | Pre_10 | 1.50±0.09 | 92.03±0.55 | 6.90±0.66 | 5.31±0.45 | 0.22±0.04 | **142.73**±10.42 |
| | Pre_20 | **1.47**±0.08 | 91.05±0.50 | 7.12±0.62 | **5.20**±0.39 | 0.22±0.03 | 145.13±9.48 |
| DFL | Pre_0 | 6.19±0.53 | 91.39±0.78 | 11.75±1.58 | 35.85±4.26 | 0.30±0.06 | 1.95±0.20 |
| | Pre_5 | 5.79±0.42 | 93.10±0.66 | 10.14±1.16 | 35.52±3.54 | 0.26±0.05 | 1.78±0.15 |
| | Pre_10 | 5.07±0.33 | 91.57±0.60 | 8.66±0.89 | 34.95±3.17 | 0.25±0.04 | **1.76**±0.14 |
| | Pre_20 | **4.75**±0.27 | **92.56**±0.54 | **7.19**±0.68 | **33.42**±2.78 | **0.22**±0.03 | 1.79±0.13 |
| BB | Pre_0 | 6.44±0.55 | 93.22±0.78 | 24.57±3.29 | 26.75±3.18 | 0.32±0.06 | 13.86±1.41 |
| | Pre_5 | 6.20±0.44 | 91.79±0.66 | 20.65±2.38 | 26.10±2.63 | 0.30±0.05 | 12.81±1.11 |
| | Pre_10 | **5.39**±0.35 | **92.90**±0.60 | 17.65±1.84 | 25.58±2.35 | 0.29±0.04 | **12.50**±0.98 |
| | Pre_20 | 5.39±0.32 | 90.65±0.54 | **14.82**±1.40 | **25.30**±2.06 | **0.25**±0.03 | 12.77±0.90 |
| NCE | Pre_0 | 1.64±0.15 | 91.67±0.84 | 13.60±1.95 | 9.98±1.27 | 0.32±0.07 | 161.73±17.47 |
| | Pre_5 | 1.58±0.12 | **92.41**±0.71 | 11.96±1.47 | 9.81±1.06 | 0.28±0.05 | 147.42±13.56 |
| | Pre_10 | 1.53±0.11 | 91.72±0.64 | 9.82±1.10 | 9.72±0.94 | 0.27±0.04 | 142.18±12.03 |
| | Pre_20 | **1.54**±0.10 | 91.11±0.58 | **8.46**±0.84 | **9.49**±0.83 | **0.24**±0.04 | **140.28**±10.56 |
| PointLTR | Pre_0 | 6.42±0.42 | **90.84**±0.60 | **6.45**±0.67 | 69.68±6.33 | 0.30±0.05 | 5.07±0.40 |
| | Pre_5 | 5.64±0.32 | 91.42±0.51 | 6.78±0.61 | 68.07±5.17 | 0.27±0.03 | 4.61±0.30 |
| | Pre_10 | 5.16±0.26 | 90.78±0.46 | 7.30±0.58 | 66.88±4.62 | 0.23±0.03 | 4.63±0.28 |
| | Pre_20 | **4.70**±0.21 | 92.72±0.41 | 7.64±0.55 | **63.58**±4.05 | **0.22**±0.02 | **4.45**±0.24 |
| PairLTR | Pre_0 | 1.53±0.09 | 93.03±0.54 | **7.80**±0.73 | 5.91±0.49 | 0.33±0.05 | 0.19±0.01 |
| | Pre_5 | 1.50±0.07 | **91.28**±0.45 | 8.21±0.64 | 5.83±0.40 | 0.29±0.03 | **0.17**±0.01 |
| | Pre_10 | 1.48±0.07 | 91.14±0.41 | 8.45±0.61 | 5.71±0.36 | 0.27±0.03 | **0.16**±0.01 |
| | Pre_20 | **1.49**±0.06 | 92.26±0.37 | 8.84±0.58 | **5.65**±0.32 | **0.26**±0.02 | **0.16**±0.01 |
| ListLTR | Pre_0 | 1.54±0.09 | 92.37±0.54 | **6.00**±0.57 | 5.79±0.48 | 0.32±0.04 | 0.17±0.01 |
| | Pre_5 | 1.53±0.08 | 92.58±0.45 | 6.41±0.50 | 5.65±0.40 | 0.27±0.03 | **0.15**±0.01 |
| | Pre_10 | 1.51±0.07 | 91.69±0.41 | 6.66±0.48 | 5.63±0.35 | 0.26±0.03 | **0.15**±0.01 |
| | Pre_20 | **1.51**±0.06 | 90.79±0.37 | 6.98±0.45 | **5.44**±0.31 | **0.24**±0.02 | **0.15**±0.01 |
| RVFE (Ours) | No Anneal | 1.51±0.06 | 91.70±0.37 | 3.79±0.24 | 6.14±0.35 | 0.24±0.02 | 43.60±2.10 |
| | Anneal | **1.46**±0.05 | **91.55**±0.29 | **3.58**±0.19 | 5.96±0.27 | **0.21**±0.02 | 42.95±1.66 |

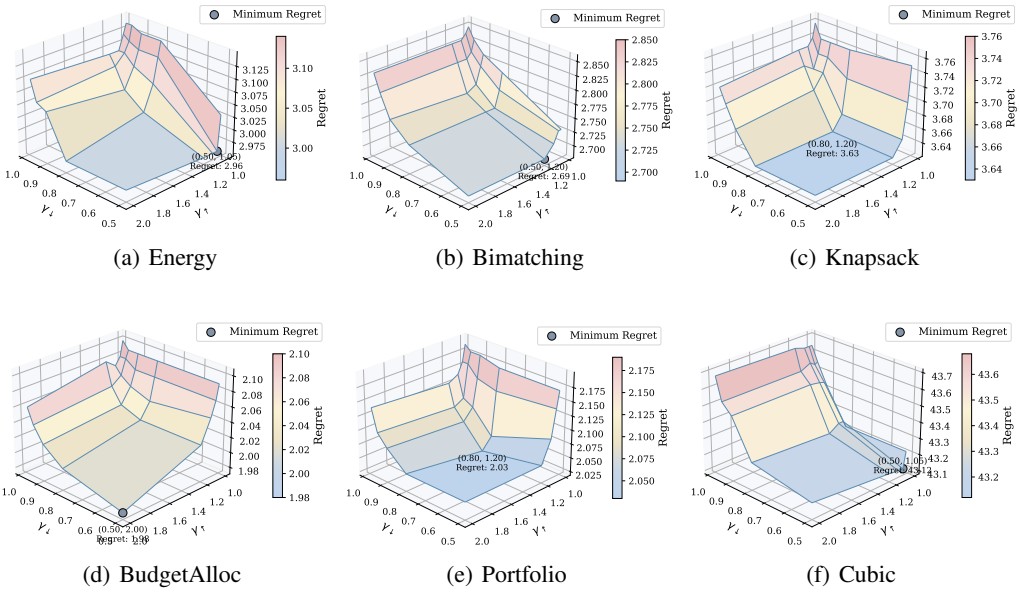

(a) Energy (b) Bimatching (c) Knapsack

(d) BudgetAlloc (e) Portfolio (f) Cubic

Figure 2: Sensitivity of RVFE to the annealing rates $(\gamma_\downarrow, \gamma_\uparrow)$ on all benchmark tasks. The regret remains low across a wide range of values, indicating RVFE's robustness to hyperparameter choices.

**Answer to Q3:** RVFE is robust to its internal hyperparameters and annealing settings. Across all six benchmarks, it maintains low regret over a broad range of annealing rates, while baseline methods exhibit much higher variability. This shows that RVFE is easier to tune and deploy in practice.

Table 2: Average training time (s, mean $\pm$ std; lower is better) per epoch for all methods across six tasks. Numbers in parentheses denote the ranking within each column (1 = fastest).

| Method | Energy | Bimatch | Knapsack | BudgetAlloc | Portfolio | Cubic |
|---|---|---|---|---|---|---|
| MSE | 0.05 ± 0.01 (1) | 0.04 ± 0.01 (1) | 0.04 ± 0.01 (1) | 0.05 ± 0.01 (1) | 0.03 ± 0.01 (1) | 0.06 ± 0.01 (1) |
| SPO+ | 2.00 ± 0.06 (5) | 0.30 ± 0.01 (4) | 0.60 ± 0.02 (5) | 4.00 ± 0.12 (4) | 0.40 ± 0.01 (6) | 0.12 ± 0.01 (5) |
| DFL | 4.00 ± 0.12 (9) | 0.80 ± 0.02 (9) | 1.50 ± 0.05 (9) | 7.00 ± 0.21 (9) | 0.60 ± 0.02 (8) | 0.16 ± 0.01 (9) |
| BB | 1.00 ± 0.03 (2) | 0.25 ± 0.01 (2) | 0.45 ± 0.01 (2) | 5.50 ± 0.17 (6) | 0.25 ± 0.01 (3) | 0.10 ± 0.01 (3) |
| NCE | 2.20 ± 0.07 (6) | 0.35 ± 0.01 (6) | 0.50 ± 0.02 (3) | 4.50 ± 0.14 (5) | 0.35 ± 0.01 (5) | 0.13 ± 0.01 (6) |
| PointLTR | 2.50 ± 0.08 (7) | 0.40 ± 0.01 (7) | 0.70 ± 0.02 (6) | 3.50 ± 0.11 (2) | 0.45 ± 0.01 (7) | 0.14 ± 0.01 (7) |
| PairLTR | 3.00 ± 0.09 (8) | 0.50 ± 0.02 (8) | 1.10 ± 0.03 (8) | 6.00 ± 0.18 (7) | 0.90 ± 0.03 (9) | 0.15 ± 0.01 (8) |
| ListLTR | 1.80 ± 0.05 (4) | 0.33 ± 0.01 (5) | 0.80 ± 0.02 (7) | 6.50 ± 0.20 (8) | 0.30 ± 0.01 (4) | 0.11 ± 0.01 (4) |
| RVFE (Ours) | 1.20 ± 0.04 (3) | 0.28 ± 0.01 (3) | 0.55 ± 0.02 (4) | 3.80 ± 0.11 (3) | 0.22 ± 0.01 (2) | 0.09 ± 0.01 (2) |

Table 3: Average testing time (s; lower is better) per evaluation for all methods across six tasks. Numbers in parentheses denote the ranking within each column (1 = fastest).

| Method | Energy | Bimatch | Knapsack | BudgetAlloc | Portfolio | Cubic |
|---|---|---|---|---|---|---|
| MSE | 11.70 (3) | 0.36 (4) | 0.65 (8) | 11.70 (2) | 0.32 (5) | 0.30 (4) |
| SPO+ | 11.50 (2) | 0.40 (6) | 0.45 (4) | 27.00 (5) | 0.32 (6) | 0.31 (5) |
| DFL | 12.30 (6) | 0.22 (2) | 0.55 (6) | 9.90 (1) | 0.27 (2) | 0.26 (2) |
| BB | 13.00 (7) | 0.30 (3) | 0.35 (1) | 46.40 (9) | 0.27 (1) | 0.27 (3) |
| NCE | 11.90 (4) | 0.38 (5) | 0.44 (3) | 28.00 (6) | 0.34 (7) | 0.32 (6) |
| PointLTR | 12.00 (5) | 1.17 (9) | 0.73 (9) | 33.70 (8) | 0.34 (8) | 0.34 (7) |
| PairLTR | 21.00 (9) | 1.05 (8) | 0.37 (2) | 31.10 (7) | 0.30 (4) | 0.55 (9) |
| ListLTR | 13.50 (8) | 0.40 (7) | 0.50 (5) | 24.30 (4) | 0.40 (9) | 0.36 (8) |
| RVFE (Ours) | 11.20 (1) | 0.15 (1) | 0.60 (7) | 16.20 (3) | 0.29 (3) | 0.25 (1) |

## 5.4 RUNTIME ANALYSIS

Tables 2 and 3 report the average training and testing time across all benchmarks. As expected, the MSE baseline is always the cheapest to train, since it never calls the downstream solver, and thus serves as a lower bound on computational cost. The more interesting comparison is among decision-focused methods. On Energy, RVFE trains in $1.20\,\mathrm{s}$ per epoch, which is comparable to BB ($1.00\,\mathrm{s}$) and clearly faster than SPO+, NCE, LTR-based losses, and generic DFL. A similar pattern appears on Bimatch and Knapsack, where RVFE keeps a top–3 training rank among decision-focused methods, and on Portfolio and Cubic it becomes the fastest decision-focused approach overall. BudgetAlloc is the only benchmark where RVFE is not the single best, yet it still stays in the faster half of decision-focused methods. The standard deviations are small (about $3\%$ of the mean), indicating stable wall-clock behavior across runs.

Testing-time costs are much closer, because all methods must solve the downstream problem at evaluation. RVFE again behaves competitively: it achieves the lowest testing time on Energy, Bimatch, and Cubic, and remains within a narrow band of the strongest baselines on the remaining tasks. Taken together with the regret results in Table 1, these observations show that RVFE does not buy accuracy by spending substantially more computation. Instead, it offers a favorable accuracy–efficiency trade-off, often matching or outperforming existing decision-focused losses while remaining among the fastest options in practice.

## 6 CONCLUSION

In this work, we revisited decision-focused learning through the lens of learning dynamics and identified instability in the decision map as a central obstacle. Our proposed recursive variational free-energy framework provides a principled way to smooth these dynamics, yielding stable training trajectories and consistently lower regret across a diverse set of benchmarks. At the same time, our empirical study reveals limitations. RVFE introduces additional hyperparameters, and while its annealing schedule improves robustness in most settings, performance can still vary across datasets. In particular, highly discontinuous tasks such as Cubic Top-$k$ expose the boundaries of smooth surrogate methods, where even RVFE cannot fully overcome structural irregularities.

## 7 ETHICS STATEMENT.

This research adheres to the ICLR Code of Ethics. Our study does not involve human subjects, personal or sensitive data, or any information that raises privacy or security concerns. The datasets used in our experiments are publicly available benchmarks, and all preprocessing steps are documented in the appendix to ensure transparency. The proposed methods focus on improving stability and reproducibility in decision-focused learning and do not introduce direct risks of harmful or unethical applications. All authors have read and agree to comply with the ICLR Code of Ethics, including standards of research integrity, fairness, and legal compliance.

## 8 REPRODUCIBILITY STATEMENT.

We have taken several steps to ensure reproducibility of our results. The main paper describes the complete problem formulations, theoretical assumptions, and algorithmic design (Sections 3–4), while all proofs are provided in the appendix for clarity and verification. Details of experimental settings, datasets, and hyperparameters are included in the appendix and supplemental materials. Together, these resources allow independent researchers to reproduce both our theoretical and empirical findings.

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

## A    USE OF LLMs

Large Language Models (LLMs) were employed to assist in the preparation of this manuscript, specifically for translation, language polishing, and correction of typographical errors. All core elements of the work were conceived, developed, and written by the human authors, including the research motivation, theoretical framework, methodological design, experimental analysis, and overall organization of the manuscript.

## B    RELATED WORK

**Decision-Focused Learning.**    Decision-focused learning (DFL) bridges prediction and optimization by training models to directly improve decision quality. This idea was formalized by (Wilder et al., 2019), who proposed integrating discrete optimization into the learning loop. Subsequent works explored differentiable solvers (Pogančić et al., 2020; Ferber et al., 2020) and surrogate loss functions that approximate decision regret, such as SPO+ (Elmachtoub & Grigas, 2022) and perturbation-based methods (Berthet et al., 2020). More recent extensions replace handcrafted surrogates with learned task-specific losses (Shah et al., 2022b), or utilize ranking-based formulations (Mandi et al., 2022; Sadana et al., 2025). Despite growing adoption, these methods may struggle when gradients are sparse or uninformative.

**Gradient Instability and Warm-Up Heuristics.**    Training a model to minimize decision loss often leads to unstable updates due to discontinuities in the decision map $w^*(\hat{c})$. Several works report that small changes in predicted parameters can lead to abrupt switches in the optimizer's output (Mulamba et al., 2021; Tang & Khalil, 2022). To mitigate this, many pipelines employ a warm-up phase: first pretraining the network on prediction loss, then fine-tuning with decision loss. While effective in practice, this approach introduces new hyperparameters and lacks theoretical guarantees (Geng et al., 2023; Mandi et al., 2020). Our work revisits this problem by studying whether or not the warm-ups are necessary.

**Learning Dynamics of Neural Networks.**    There is growing recognition that training dynamics—especially around non-smooth regions—play a critical role in deep learning performance (Kolter & Manek, 2019; Jiang et al., 2024; Zhao et al., 2024). In machine learning, RVFE has been applied to model neural dynamics (Sidky & Whitmer, 2018) and to build energy-based solvers for optimization tasks (Shen et al., 2025). Our work takes advantage of this perspective to model DFL training as a dynamic system navigating discontinuous decision landscapes.

**Adaptive Training and Annealing Methods.**    Several methods have been proposed to stabilize learning in decision-focused settings. Annealing approaches gradually increase the weight of the decision loss, smoothing the transition from prediction to optimization (Pogančić et al., 2020). Other strategies use black-box solvers with projected gradients (Sahoo et al.), or sample perturbations to estimate gradients in non-differentiable regimes (Niepert et al., 2021). While effective, these techniques often rely on fixed schedules or tuning and lack a unifying view of the underlying dynamics.

## C   DETAILED PROOFS

### C.1   PROOF OF THEOREM 3.1

*Proof.* Fix a prediction $\hat{c}_0 \in \mathbb{R}^d$ and a unit direction $v \in \mathbb{R}^d$. Recall $\widetilde{\Delta} := (\sigma_{\text{pred}}/\sigma_{\text{dec}})\,\Delta$ with $\Delta := v^\top(w_a - w_b)$, and define the signed gap

$$B := v^\top(\hat{c}_0 - c) - \tfrac{1}{2}\widetilde{\Delta}.$$

For $\epsilon > 0$ let $\hat{c}_\epsilon := \hat{c}_0 + \epsilon v$.

**(a) Directional derivative of the prediction loss.**   Using the quadratic expansion,

$$L_{\text{pred}}(\hat{c}_\epsilon, c) = \frac{\|(\hat{c}_0 - c) + \epsilon v\|^2}{\sigma_{\text{pred}}} = \frac{\|\hat{c}_0 - c\|^2}{\sigma_{\text{pred}}} + \frac{2\epsilon}{\sigma_{\text{pred}}}\,v^\top(\hat{c}_0 - c) + O(\epsilon^2),$$

so

$$\partial_v L_{\text{pred}}(\hat{c}_0, c) = \lim_{\epsilon \downarrow 0} \frac{L_{\text{pred}}(\hat{c}_\epsilon, c) - L_{\text{pred}}(\hat{c}_0, c)}{\epsilon} = \frac{2}{\sigma_{\text{pred}}}\,v^\top(\hat{c}_0 - c).$$

**(b) Directional derivative of the decision loss.**   Let $\mathcal{W} \subset \mathbb{R}^d$ be a finite nonempty set. For $x \in \mathbb{R}^d$ define the value function

$$\phi(x) := \min_{w \in \mathcal{W}} x^\top w,$$

and select an optimal decision $w^\star(x) \in \arg\min_{w \in \mathcal{W}} x^\top w$ according to a fixed tie-breaking rule. Consider a boundary point $x_0 \in \mathbb{R}^d$ at which there are exactly two active minimizers $w_a, w_b \in \mathcal{W}$ such that

$$x_0^\top w_a = x_0^\top w_b = \min_{w \in \mathcal{W}} x_0^\top w,$$

and $x_0^\top w > x_0^\top w_a$ for all $w \in \mathcal{W} \setminus \{w_a, w_b\}$. Set $d := w_a - w_b$.

Fix a direction $v \in \mathbb{R}^d$ and define $\Delta := v^\top d$. For $\epsilon > 0$ we have

$$(x_0 + \epsilon v)^\top w_a - (x_0 + \epsilon v)^\top w_b = \epsilon\, v^\top(w_a - w_b) = \epsilon\,\Delta.$$

If $\Delta > 0$, then for all sufficiently small $\epsilon > 0$,

$$(x_0 + \epsilon v)^\top w_b < (x_0 + \epsilon v)^\top w_a \quad \text{and} \quad (x_0 + \epsilon v)^\top w_b < (x_0 + \epsilon v)^\top w \ \ \forall w \in \mathcal{W} \setminus \{w_a, w_b\},$$

so $w^\star(x_0 + \epsilon v) = w_b$ and

$$\phi(x_0 + \epsilon v) = (x_0 + \epsilon v)^\top w_b = x_0^\top w_b + \epsilon\, v^\top w_b.$$

Since $\phi(x_0) = x_0^\top w_b$ holds as well, it follows that

$$\partial_v^+ \phi(x_0) := \lim_{\epsilon \downarrow 0} \frac{\phi(x_0 + \epsilon v) - \phi(x_0)}{\epsilon} = v^\top w_b \qquad \text{when } \Delta > 0.$$

If $\Delta < 0$, the roles of $w_a$ and $w_b$ are interchanged. For all sufficiently small $\epsilon > 0$ we have

$$(x_0 + \epsilon v)^\top w_a < (x_0 + \epsilon v)^\top w_b \quad \text{and} \quad (x_0 + \epsilon v)^\top w_a < (x_0 + \epsilon v)^\top w \ \ \forall w \in \mathcal{W} \setminus \{w_a, w_b\},$$

so $w^\star(x_0 + \epsilon v) = w_a$ and

$$\phi(x_0 + \epsilon v) = (x_0 + \epsilon v)^\top w_a = x_0^\top w_a + \epsilon\, v^\top w_a.$$

Using $\phi(x_0) = x_0^\top w_a$ in this case, we obtain

$$\partial_v^+ \phi(x_0) = v^\top w_a \qquad \text{when } \Delta < 0.$$

Thus, at a boundary point where $w_a$ and $w_b$ exchange optimality, the one-sided directional derivative of the value function satisfies

$$\partial_v^+ \phi(x_0) = \begin{cases} v^\top w_b, & v^\top(w_a - w_b) > 0, \\ v^\top w_a, & v^\top(w_a - w_b) < 0. \end{cases}$$

The decision loss used in Theorem 3.1 is obtained from $\phi$ by a linear rescaling and an additive constant (independent of the decision argument), so its one-sided directional derivative inherits the same sign pattern.

**(c) Lyapunov derivative and choice of the weight.** For any $\lambda > 0$,

$$\partial_v V_\lambda(\hat{c}_0) = \lambda \, \partial_v L_{\mathrm{pred}}(\hat{c}_0, c) + \partial_v^+ L_{\mathrm{dec}}(\hat{c}_0, c) = \frac{2\lambda}{\sigma_{\mathrm{pred}}} \, v^\top(\hat{c}_0 - c) - \frac{\Delta}{\sigma_{\mathrm{dec}}} \, \mathrm{sgn}(B).$$

Using $v^\top(\hat{c}_0 - c) = B + \frac{1}{2}\widetilde{\Delta}$ and $\widetilde{\Delta} = (\sigma_{\mathrm{pred}}/\sigma_{\mathrm{dec}})\Delta$,

$$\partial_v V_\lambda(\hat{c}_0) = \frac{2\lambda}{\sigma_{\mathrm{pred}}}\Big(B + \tfrac{1}{2}\widetilde{\Delta}\Big) - \frac{\Delta}{\sigma_{\mathrm{dec}}} \, \mathrm{sgn}(B) = \frac{2\lambda}{\sigma_{\mathrm{pred}}} \, B + \Big(\frac{\lambda}{\sigma_{\mathrm{pred}}}\widetilde{\Delta} - \frac{\Delta}{\sigma_{\mathrm{dec}}} \, \mathrm{sgn}(B)\Big).$$

Choose

$$\lambda^\star = \frac{\sigma_{\mathrm{pred}}}{2\sigma_{\mathrm{dec}}}\Big(1 + \frac{\widetilde{\Delta}}{2B}\Big) \, \mathrm{sgn}(B) \quad \text{for } B \neq 0,$$

which satisfies $\lambda^\star > 0$ and yields

$$\partial_v V_{\lambda^\star}(\hat{c}_0) = 2B \, \mathrm{sgn}(B).$$

**(d) Separation hyperplane.** The set

$$\mathcal{H}_{\mathrm{scaled}} = \Big\{\hat{c} \in \mathbb{R}^d \; : \; v^\top(\hat{c} - c) = \tfrac{1}{2}\widetilde{\Delta}\Big\}$$

is precisely $\{B = 0\}$. If $B > 0$, then $\partial_v L_{\mathrm{pred}}(\hat{c}_0, c) > 0$ and $\partial_v^+ L_{\mathrm{dec}}(\hat{c}_0, c) < 0$, so moving along $+v$ decreases both losses (and $\partial_v V_{\lambda^\star}(\hat{c}_0) < 0$). If $B < 0$, the two directional derivatives have opposite signs and $\partial_v V_{\lambda^\star}(\hat{c}_0) > 0$. Therefore $\mathcal{H}_{\mathrm{scaled}}$ separates the stable and trade-off regions, completing the proof. $\qquad\square$

## C.2 PROOF OF PROPOSITION 4.1

*Proof.* Decompose the objective:

$$\frac{1}{\beta}\Big(\frac{1}{\sigma_{\mathrm{dec}}}\mathbb{E}_q[c^\top w] + \mathrm{KL}(q \,\|\, P_\beta)\Big) = \frac{1}{\beta}\sum_w q(w)\Big(\frac{1}{\sigma_{\mathrm{dec}}}c^\top w + \log\frac{q(w)}{P_\beta(w \mid \hat{c})}\Big).$$

For fixed $(\hat{c}, \beta)$, this is a convex functional with respect to $q$.

Point-wise minimization gives

$$q^\star(w) \propto \exp\big(-c^\top w/\sigma_{\mathrm{dec}}\big)P_\beta(w \mid \hat{c}).$$

Consider that $P_\beta(\cdot \mid \hat{c})$ is already a probability measure, the proportionality constant is one iff

$$q^\star = P_\beta(\cdot \mid \hat{c})$$

.

Substituting $q^\star$ back eliminates the KL term and yields $\frac{1}{\beta\sigma_{\mathrm{dec}}} c^\top \bar{w}_\beta(\hat{c})$, which together with the MSE and the constant $-c^\top w^*(c)/(\beta\sigma_{\mathrm{dec}})$ gives the stated minimum value. $\qquad\square$

## C.3 PROOF OF PROPOSITION 4.2

*Proof.* For the prediction term, $\langle \nabla_{\hat{c}}\|\hat{c} - c\|^2/\sigma_{\mathrm{pred}}, \, d\rangle = (2/\sigma_{\mathrm{pred}})\langle \hat{c} - c, d\rangle = 0$ because $d$ lies in the *decision* space.

For $L_{\mathrm{dec}}^{\mathrm{soft}}$ the soft-max gradient identity gives

$$\nabla_{\hat{c}} L_{\mathrm{dec}}^{\mathrm{soft}} = -\frac{\tau}{\beta\sigma_{\mathrm{dec}}} \, \mathrm{Cov}_{P_\tau}(w, c^\top w).$$

Its inner product with $d$ has *unknown* sign but is finite.

Using $\delta_{\mu,\tau}(w) = \mu\|w - \bar{w}\|_2$,

$$\nabla_{\hat{c}}\frac{1}{\beta\sigma_{\mathrm{dec}}}\mathbb{E}_{P_\tau}\big[\delta_{\mu,\tau}(w)\big] = -\frac{\tau\mu}{\beta\sigma_{\mathrm{dec}}} \, \mathrm{Cov}_{P_\tau}\big(w, \|w - \bar{w}\|\big).$$

Project onto $d$:
$$\langle \mathrm{Cov}(w, \|w - \bar{w}\|),\, d \rangle = \mathrm{Cov}\big(\langle w, d \rangle,\, \|w - \bar{w}\|\big).$$

Both terms inside Cov are *strictly monotone* in $\langle w, d \rangle$, so the covariance equals $\mathrm{Var}_{P_\tau}(\langle w, d \rangle)$ and is positive whenever the variance is non-zero. Hence
$$\big\langle \nabla_{\hat{c}} \mathcal{F}_{\tau,\mu},\, d \big\rangle = -\tfrac{\tau\mu}{\beta\sigma_{\mathrm{dec}}} \mathrm{Var}_{P_\tau}(\langle w, d \rangle) + \underbrace{\langle (ii), d \rangle}_{\text{finite}} < 0.$$

Since the variance term is strictly positive, the whole inner product is negative. $\qquad\square$

### C.4 PROOF OF PROPOSITION 4.4

*Proof.* Let $X := w \in \mathbb{R}^d$ and $Y := f(w) \in \mathbb{R}$. Write population means $\mu_X := \mathbb{E}_{P_\tau}[X] = \bar{w}$, $\mu_Y := \mathbb{E}_{P_\tau}[Y] = \bar{f}$, and the vector covariance $\Sigma_{XY} := \mathrm{Cov}_{P_\tau}(X, Y) = \mathbb{E}[(X - \mu_X)(Y - \mu_Y)]$. Then $g(\hat{c}) = (2/\sigma_{\mathrm{pred}})(\hat{c} - c) + (\tau/(\beta\sigma_{\mathrm{dec}}))\,\Sigma_{XY}$.

Let $(X_k, Y_k)$ be i.i.d. from $(X, Y)$ under $P_\tau$. Define $\bar{X} := \frac{1}{K}\sum_k X_k$, $\bar{Y} := \frac{1}{K}\sum_k Y_k$. Use the identity
$$\sum_{k=1}^{K}(X_k - \bar{X})(Y_k - \bar{Y}) = \sum_{k=1}^{K} X_k Y_k - K\,\bar{X}\bar{Y}.$$

Taking expectations and using independence,
$$\mathbb{E}\left[\sum_{k=1}^{K} X_k Y_k\right] = K\,\mathbb{E}[X_1 Y_1], \qquad \mathbb{E}[\bar{X}\bar{Y}] = \frac{1}{K}\mathbb{E}[X_1 Y_1] + \frac{K-1}{K}\mu_X\mu_Y.$$

Therefore
$$\mathbb{E}\left[\sum_{k=1}^{K}(X_k - \bar{X})(Y_k - \bar{Y})\right] = (K-1)\big(\mathbb{E}[X_1 Y_1] - \mu_X\mu_Y\big) = (K-1)\,\Sigma_{XY},$$

and dividing by $K - 1$ gives $\mathbb{E}[\widehat{\mathrm{Cov}}(w, f)] = \Sigma_{XY}$.

The term $(2/\sigma_{\mathrm{pred}})(\hat{c} - c)$ is deterministic, so
$$\mathbb{E}[\widehat{g}(\hat{c})] = \frac{2}{\sigma_{\mathrm{pred}}}(\hat{c} - c) + \frac{\tau}{\beta\sigma_{\mathrm{dec}}}\mathbb{E}[\widehat{\mathrm{Cov}}(w, f)] = \frac{2}{\sigma_{\mathrm{pred}}}(\hat{c} - c) + \frac{\tau}{\beta\sigma_{\mathrm{dec}}}\Sigma_{XY} = g(\hat{c}).$$

For variance, consider $\widetilde{\Sigma}_{XY} := \frac{1}{K}\sum_{k=1}^{K}(X_k - \mu_X)(Y_k - \mu_Y)$. Then $\mathrm{Var}(\widetilde{\Sigma}_{XY}) = \mathrm{Var}\big((X_1 - \mu_X)(Y_1 - \mu_Y)\big)/K = \mathcal{O}(1/K)$ componentwise by the finite second moments. A standard decomposition shows $\widehat{\Sigma}_{XY} - \widetilde{\Sigma}_{XY} = \frac{1}{K}\sum_{k=1}^{K}\big[(\mu_X - \bar{X})(Y_k - \mu_Y) + (X_k - \mu_X)(\mu_Y - \bar{Y})\big] + \frac{K}{K-1}(\bar{X} - \mu_X)(\bar{Y} - \mu_Y)$, whose variance is $\mathcal{O}(1/K^2)$. Hence $\mathrm{Var}(\widehat{\Sigma}_{XY}) = \mathcal{O}(1/K)$ componentwise, and multiplying by $(\tau/(\beta\sigma_{\mathrm{dec}}))^2$ yields $\mathrm{Var}(\widehat{g}(\hat{c})) = \mathcal{O}(1/K)$ componentwise.

Let $\Phi(w) := w\,f(w) \in \mathbb{R}^d$. Assume $|w_j| \leq 1$ and $\|f\|_\infty \leq M$, so $|\Phi_j(w)| \leq M$ for each coordinate $j$. By the total variation dual bound, for any bounded scalar $\phi$, $|\mathbb{E}_Q[\phi] - \mathbb{E}_P[\phi]| \leq 2\|\phi\|_\infty \mathrm{TV}(Q, P)$. Apply this to each coordinate $\Phi_j$ to obtain
$$\big|\mathbb{E}_{Q_\tau}[\Phi_j] - \mathbb{E}_{P_\tau}[\Phi_j]\big| \leq 2M\,\delta, \qquad j = 1, \ldots, d,$$

hence $\|\mathbb{E}_{Q_\tau}[w f(w)] - \mathbb{E}_{P_\tau}[w f(w)]\|_2 \leq 2M\,\delta$.

Write $\mathrm{Cov}(w, f) = \mathbb{E}[wf] - \mathbb{E}[w]\mathbb{E}[f]$. The same TV bound with $\phi(w) = w_j$ (bounded by 1) and $\phi(w) = f(w)$ (bounded by $M$) gives
$$\|\mathbb{E}_{Q_\tau}[w] - \mathbb{E}_{P_\tau}[w]\|_\infty \leq 2\,\delta, \qquad |\mathbb{E}_{Q_\tau}[f] - \mathbb{E}_{P_\tau}[f]| \leq 2M\,\delta.$$

For each coordinate $j$,
$$\big|\mathrm{Cov}_{Q_\tau}(w, f)_j - \mathrm{Cov}_{P_\tau}(w, f)_j\big| \leq \big|\mathbb{E}_{Q_\tau}[w_j f] - \mathbb{E}_{P_\tau}[w_j f]\big| + \big|\mathbb{E}_{Q_\tau}[w_j]\mathbb{E}_{Q_\tau}[f] - \mathbb{E}_{P_\tau}[w_j]\mathbb{E}_{P_\tau}[f]\big|$$
$$\leq 2M\,\delta + \big|\mathbb{E}_{Q_\tau}[w_j] - \mathbb{E}_{P_\tau}[w_j]\big| \cdot |\mathbb{E}_{Q_\tau}[f]| + \big|\mathbb{E}_{Q_\tau}[f] - \mathbb{E}_{P_\tau}[f]\big| \cdot |\mathbb{E}_{P_\tau}[w_j]|$$
$$\leq 2M\,\delta + (2\delta) \cdot M + (2M\,\delta) \cdot 1 = 6M\,\delta.$$

Thus componentwise $\|\mathrm{Cov}_{Q_\tau}(w, f) - \mathrm{Cov}_{P_\tau}(w, f)\|_\infty \leq 6M\,\delta$, and therefore $\|\mathrm{Cov}_{Q_\tau}(w, f) - \mathrm{Cov}_{P_\tau}(w, f)\|_2 \leq 6M\,\delta$. Multiplying by $\tau/(\beta\sigma_{\mathrm{dec}})$ gives the stated gradient bias bound.

Estimating expectations with $K$ i.i.d. draws from $Q_\tau$ yields the same $\mathcal{O}(1/K)$ variance decay componentwise as in Step 3 (the argument only uses i.i.d. sampling and finite second moments). Combining bias and variance gives the stated mean-square error bound. $\qquad\square$

## C.5 Proof of Theorem 4.5

*Proof.* We prove that the free energy $\mathcal{F}_{\mathrm{soft}}(\hat{c}; \beta) := \|\hat{c} - c\|^2/\sigma_{\mathrm{pred}} + L_{\mathrm{dec}}^{\mathrm{soft}}(\hat{c}, c; \beta)$ decreases along one iteration of gradient descent whenever $B_t \geq 0$ (stable half-space).

Fix an iteration $t$ with current prediction $\hat{c}_t$ and temperature $\beta_t$. Write $g_t := \nabla_{\hat{c}}\mathcal{F}_{\mathrm{soft}}(\hat{c}_t; \beta_t)$ for the full gradient and denote its Lipschitz constant on the current decision cell by $L_t > 0$ (smoothness follows from the polyhedral structure plus the softmax surrogate).

Choose a step size $\eta_t \leq 1/L_t$ and update

$$\hat{c}_{t+1} := \hat{c}_t - \eta_t g_t.$$

Because $\hat{c}_t$ stays inside the same cell for sufficiently small $\eta_t$ (the facet is codimension 1), the classical *descent lemma* for $L_t$-smooth functions applies:

$$\mathcal{F}_{\mathrm{soft}}(\hat{c}_{t+1}; \beta_t) \leq \mathcal{F}_{\mathrm{soft}}(\hat{c}_t; \beta_t) - \frac{\eta_t}{2}\|g_t\|^2. \tag{D1}$$

Since $B_t \geq 0$ the adaptive schedule sets $\beta_{t+1} = \gamma_\downarrow \beta_t$ with $0 < \gamma_\downarrow < 1$; hence $\beta_{t+1} < \beta_t$. For *fixed* $\hat{c}$ the soft decision loss satisfies

$$L_{\mathrm{dec}}^{\mathrm{soft}}(\hat{c}, c; \beta_{t+1}) = \gamma_\downarrow^{-1} L_{\mathrm{dec}}^{\mathrm{soft}}(\hat{c}, c; \beta_t) \quad \Longrightarrow \quad L_{\mathrm{dec}}^{\mathrm{soft}}(\hat{c}, c; \beta_{t+1}) < L_{\mathrm{dec}}^{\mathrm{soft}}(\hat{c}, c; \beta_t).$$

Therefore, at the same point $\hat{c}_{t+1}$,

$$\mathcal{F}_{\mathrm{soft}}(\hat{c}_{t+1}; \beta_{t+1}) \leq \mathcal{F}_{\mathrm{soft}}(\hat{c}_{t+1}; \beta_t). \tag{D2}$$

Chaining the two inequalities yields

$$\boxed{\mathcal{F}_{\mathrm{soft}}(\hat{c}_{t+1}; \beta_{t+1}) \ \leq \ \mathcal{F}_{\mathrm{soft}}(\hat{c}_t; \beta_t) - \frac{\eta_t}{2}\|g_t\|^2}.$$

Hence the free energy strictly decreases whenever $\|g_t\| > 0$. Summing the telescoping series over all "stable" iterations gives

$$\sum_{t:\, B_t \geq 0} \eta_t \|g_t\|^2 \ \leq \ 2\,\mathcal{F}_{\mathrm{soft}}(\hat{c}_0; \beta_0) < \infty,$$

so $\|g_t\| \to 0$ along a subsequence, and because $\eta_t$ stays bounded away from zero the entire sequence converges to a first-order stationary point lying in the stable half-space. $\qquad\square$

# D Benchmark Problem

## D.1 Energy-cost aware Scheduling

With the increasing integration of clean and renewable energy sources, energy demand and market prices have become more dynamic to external factors such as weather and production forecasts (). In industrial settings, this variability opens up opportunities for energy-aware scheduling. The task allocation is optimized with respect to real-time or forecasted energy prices—leading to substantial reductions in energy consumption and operational expenditure.

**Prediction Task:** The upstream task is to forecast energy prices over $T = 48$ half-hour time slots per day. The input feature vector for each slot includes calendar-based indicators, weather forecasts, day-ahead estimates of wind energy production, energy load and pricing, as well as actual wind speed, temperature, $CO_2$ intensity.

**Optimization Task:** The downstream task is to schedule $J$ jobs on $M$ machines to minimize the total energy cost, subject to temporal and resource constraints. Each job $j \in J$ is associated with an earliest start time $e_j$, latest finish time $l_j$, duration $d_j$, power usage $p_j$, and resource usage $u_{jr}$. Each machine $m \in M$ has capacity $c_{mr}$ for resource $r \in R$. Let $\mathbf{z}^{jmt} \in \{0,1\}$ indicate whether job $j$ starts on machine $m$ at time $t$. The cost-minimizing scheduling problem is formulated as:

$$\min_{\mathbf{z}^{jmt}} \quad \sum_{j \in J} \sum_{m \in M} \sum_{t \in T} \mathbf{z}^{jmt} \left( \sum_{t \le t' < t + d_j} p_j \mathbf{y}^{t'} \right) \tag{2}$$

$$\tag{3}$$

$$\text{s.t.} \quad \sum_{m \in M} \sum_{t \in T} \mathbf{z}^{jmt} = 1, \qquad\qquad \forall j \in J \tag{4}$$

$$\tag{5}$$

$$\mathbf{z}^{jmt} = 0, \qquad\qquad \forall j, m, t < e_j \tag{6}$$

$$\tag{7}$$

$$\mathbf{z}^{jmt} = 0, \qquad\qquad \forall j, m, t + d_j > l_j \tag{8}$$

$$\tag{9}$$

$$\sum_{j \in J} \sum_{t - d_j < t' \le t} \mathbf{z}^{jmt'} u_{jr} \le y^{mr}, \qquad\qquad \forall m \in M, \forall r \in R, \forall t \in T \tag{10}$$

Constraint equation 4 ensures each job is scheduled exactly once. Constraints equation 6 and equation 8 enforce that scheduling respects temporal bounds. Constraint equation 10 ensures that total resource usage at any time does not exceed the machine's capacity.

**Dataset:** We use open-access data from the Irish Single Electricity Market Operator (SEMO)(Ifrim et al., 2012). SEMO covers the period from November 1st, 2011 to December 31st, 2013. Each day is treated as an independent optimization instance. Our experiments use $M = 3$ machines and $R = 1$ resource. The resource usage $u_{jr}$ is deterministic and known.

## D.2 KNAPSACK

The knapsack problem with uncertain rewards appears in many real-world tasks. One example is when an agent must select items to carry, aiming to reduce energy or transport costs. In these settings, the item values (e.g., energy cost savings) are unknown and must be predicted.

**Prediction Task:** The goal is to predict the value $\mathbf{y}^j$ of each item $j$, given its feature vector $\mathbf{x}^j$, for $N$ items.

**Optimization Task:** Given the predicted values, the decision problem is to select items that maximize total value without exceeding the capacity constraint. The problem is formulated as:

$$\mathbf{z}^{\star}(\mathbf{y}) = \arg\max_{\mathbf{z}} \sum_{j=1}^{N} \mathbf{y}^j \mathbf{z}^j \quad \text{s.t.} \quad \sum_{j=1}^{N} \mathbf{w}^j \mathbf{z}^j \le C,$$

where $\mathbf{z}^j \in \{0,1\}$ indicates whether item $j$ is selected, $\mathbf{w}^j$ is its weight, and $C$ is the total knapsack capacity.

**Dataset.** We generate $n$ samples $\{(\mathbf{x}_i, \mathbf{y}_i)\}_{i=1}^{n}$ using a polynomial model(Tang & Khalil, 2022):

$$\mathbf{y}_i = \left[ \frac{1}{3.5^{\deg} \sqrt{p}} \left( \mathcal{B} \mathbf{x}_i + 3 \right)^{\deg} + 1 \right] \cdot \epsilon_i,$$

where $\mathbf{x}_i \sim \mathcal{N}(0, I_p)$ is a standard Gaussian input, and $\mathcal{B} \in \mathbb{R}^{d \times p}$ is the ground-truth coefficient matrix with Bernoulli(0.5) entries. The noise term $\epsilon_i^j$ is sampled uniformly. The number of features is $p$. The item weights $\mathbf{w}^j$ are sampled uniformly between 3 and 8. We set the capacity $C = 30$, the number of items $N = 20$, and the polynomial degree $\deg = 4$.

### D.3  BUDGE ALLOCATION UNDER UNCERTAINTY

The Budge Allocation problem models situations where organizations want to spread information across several websites with limited budget. For example, a nonprofit may aim to reach as many users as possible by buying space on a small number of sites.

**Prediction Task:** For each website $w$, we are given features $\mathbf{x}^w$. The task is to predict $\mathbf{y}^{wu}$, the probability that user $u$ sees the content through website $w$.

**Optimization Task:** The optimization goal is to choose which websites to use under a budget limit. The objective is to maximize the expected number of users reached at least once, of which is defined as:

$$\mathbf{z}^\star(\mathbf{y}) = \arg\max_{\mathbf{z}} \frac{1}{N} \sum_{u=1}^{N} \left( 1 - \prod_{w=1}^{M} (1 - \mathbf{z}^w \cdot \mathbf{y}^{wu}) \right) \quad \text{s.t.} \quad \sum_{w=1}^{M} \mathbf{z}^w \leq B,$$

where $\mathbf{z}^w \in \{0, 1\}$ selects website $w$, $N$ is the number of users, and $M$ is the number of websites.

**Dataset:** We use the Yahoo! Webscope dataset with user-level labels $\mathbf{y}_i$. Following prior work(Wilder et al., 2019), we generate features by multiplying a random matrix $\mathbf{A} \in \mathbb{R}^{N \times N}$ with $\mathbf{y}_i$, making sure that $\mathbf{x}_i = \mathbf{A}\mathbf{y}_i$. In our setup, we use $M = 5$ websites, $N = 10$ users, and a budget $B = 1$, consistent with(Shah et al., 2022b).

### D.4  BIPARTITE MATCHING

Graph matching is widely used in social networks, helping users find connections like friends or job opportunities. However, the links between nodes are not always known. In such cases, we must first predict the edges before performing matching. This setup follows prior work(Mandi et al., 2022).

**Prediction Task:** Given a pair of nodes $(i, j)$, we use their features $\mathbf{x}^i$ and $\mathbf{x}^j$ to predict whether an edge exists. Let $\mathbf{y}^{ij} = 1$ if there is a link, and $\mathbf{y}^{ij} = 0$ otherwise. The prediction model learns:

$$\mathbf{y}^{ij} = \mathcal{M}([\mathbf{x}^i, \mathbf{x}^j]).$$

**Optimization Task:** Given predicted edges, the matching problem selects node pairs to maximize total match score. The solution $\mathbf{z}^{ij} = 1$ means node $i$ is matched to node $j$. The formulation is:

$$\mathbf{z}^\star(\mathbf{y}) = \arg\max_{\mathbf{z}} \sum_{i=1}^{N} \sum_{j=1}^{N} \mathbf{y}^{ij} \mathbf{z}^{ij} \quad \text{s.t.} \quad \mathbf{z}\mathbf{1} = \mathbf{1}, \quad \mathbf{z}^\top \mathbf{1} = \mathbf{1},$$

where $\mathbf{y} \in \mathbb{R}^{N \times N}$ is the predicted adjacency matrix, and $\mathbf{z} \in \mathbb{R}^{N \times N}$ is the assignment matrix under permutation constraints.

**Dataset:** Following prior research(Wilder et al., 2019), we use the Cora citation network, where each node is a paper with a 1433-dimensional feature vector from bag-of-words.

### D.5  PORTFOLIO

Asset allocation is a key tool for moving capital efficiently across sectors and improving overall economic performance.

**Prediction Task.** Given historical features such as daily prices and trading volumes, the goal is to predict the next-day return $\mathbf{y}^i$ for each of the $N$ stocks.

**Optimization Task.** The objective is to choose investment weights that balance return and risk. We use the classical Markowitz formulation(Geng et al., 2023):

$$\mathbf{z}^\star(\mathbf{y}) = \arg\max_{\mathbf{z}} \ \mathbf{z}^\top \mathbf{y} - \lambda \mathbf{z}^\top \mathbf{Q}\mathbf{z} \quad \text{s.t.} \quad \sum_{i=1}^{N} \mathbf{z}^i = 1,$$

where $\mathbf{z}^i \in [0, 1]$ is the fraction of capital invested in stock $i$, $\lambda = 0.1$ is the risk-aversion parameter, and $\mathbf{Q} \in \mathbb{R}^{N \times N}$ is the covariance matrix of returns.

Table 4: Summary of datasets and optimization problems

| Dataset | Instances (Train/Test) | Variable Dim. | Problem Type | Solver |
|---|---|---|---|---|
| Energy | 650 / 139 | 48 | ILP | Gurobi(Gurobi Optimization, LLC, 2024) |
| Knapsack | 400 / 200 | 20 | ILP | Gurobi(Gurobi Optimization, LLC, 2024) |
| Cubic | 250 / 400 | 50 | Top-$k$ | Heuristic |
| BudgetAlloc | 400 / 200 | 100 | Neural | Neural Solver(Karimi et al., 2017) |
| BiMatching | 200 / 6 | 50 | ILP | CVXPY(Diamond & Boyd, 2016) |
| Portfolio | 400 / 200 | 50 | QP | CVXPY(Diamond & Boyd, 2016) |

**Dataset.** We use the S&P 500 dataset. Features are constructed from past returns over different time windows—10 days, weeks, months, and years—as well as their rolling averages. The risk aversion parameter is set to $\lambda = 0.1$.

### D.6 Cubic Top-$k$

The Top-$k$ selection problem is widely used in explainable machine learning, where identifying the most relevant features or elements is the core goal.

**Prediction Task.** Given a dataset of feature–target pairs $\{\mathbf{x}_i, y_i\}$, each input $x_i$ is sampled from a uniform distribution $x_i \sim \mathcal{U}(0, 1)$. The corresponding output is generated by a cubic polynomial:

$$y_i = 10x_i^3 - 6.5x_i.$$

**Optimization Task.** The aim is to select the top-$k$ entries with the highest predicted values:

$$\mathbf{z}^\star(\mathbf{y}) = \arg \max_{\text{top-}k} \mathbf{y}.$$

We set the total number of items to $N = 50$ and the selection budget to $k = 5$. This synthetic task is constructed from scratch and does not rely on external datasets(Shah et al., 2022b).

## E  Detailed Experimental Settings

### E.1  Prediction model

For a fair comparison, all end-to-end training models, as well as the two-stage approach, utilize the same multi-layer perception (MLP) for the prediction of optimization coefficients. The prediction model $\mathcal{M}$ using MLP is $\mathbf{a}^{(i+1)} = \sigma\left(\mathbf{W}^{(i)}\mathbf{a}^{(i)} + \mathbf{b}^{(i)}\right),\quad i = 1, 2, \ldots, K-1$, where $\mathbf{a}^{(1)} = \mathbf{x}$ and $\mathbf{y} = \mathbf{a}^{(K)}$ are input and output for $\mathcal{M}$, $a^i$ is the hidden vector for $i = 2, \cdots, K-1$, $\mathbf{W}$ is the weight term, $b$ is the bias term and $\sigma$ is the activation function where we adopt ReLU. In experiments, we adopt $K = 3$ layers and the size of intermediate hidden units is set as 32 .

### E.2  Datasets

We evaluate RVFE across a diverse suite of real and synthetic datasets (Geng et al., 2023). These tasks span different types of combinatorial and continuous optimization problems, including integer programs, submodular maximization, and quadratic programming. Table 4 summarizes the setup for each benchmark, including data source, input dimensions, train/test split, optimization type, and solver used.

### E.3  Model Architectures and Training Details

We use a two-layer fully-connected neural network with 32 hidden units and ReLU activation. Mean pooling is applied to the hidden representations, with a kernel size of 1. Training is conducted using the Adam optimizer with a learning rate of $1 \times 10^{-2}$, batch size of 256, and a maximum of 100 epochs. Early stopping is used with a patience of 10 epochs.

Table 5: Experimental hyperparameters

| Parameter | Value |
| --- | --- |
| Learning Rate | $1 \times 10^{-2}$ |
| Batch Size | 256 |
| Epochs | 100 (early stopping: patience = 10) |
| Pretraining Epochs | 0, 5, 10, 20 |
| Solver | Gurobi 12.0 (exact MILP/LP) |
| Random Seed | 2025 |
| GPU | NVIDIA RTX 4080 |
| CPU | Intel Core i7-13900K |
| System Memory | 64 GB RAM |
| Framework | PyTorch 1.13 |

We solve the downstream optimization problem using Gurobi 12.0 to obtain exact MILP/LP solutions. All experiments are executed on a machine equipped with an NVIDIA RTX 4080 GPU, an Intel Core i7-13900K CPU, and 64 GB of RAM. The implementation is based on PyTorch 1.13.

Table 5 summarizes the hyperparameters used in all experiments. We vary the number of pretraining epochs over 0, 5, 10, 20, and tune the annealing parameters $\beta \in \{0.01, 0.05, 0.1\}$, with $\tau$ adjusted dynamically throughout training. All runs use a fixed random seed for reproducibility.

# F PRETRAINING AND REGRET: HOW STABLE ARE EXISTING METHODS?

To evaluate the necessity and impact of pretraining in decision-focused methods, we vary the number of pretraining steps across a range of decision baselines. Figure 3 shows regret values under different pretraining lengths (0 to 20 steps) across six benchmarks.

As the plots show, all baseline methods—SPO+, DFL, BB, NCE, and LTR variants—show noticeable variance across pretraining levels. Some methods (e.g., BB in Knapsack or Cubic) perform poorly without sufficient warm-up. Even with tuning, performance can remain far from optimal. In contrast, our method, RVFE with annealing, achieves a consistently low regret without requiring pretraining (flat orange line). This demonstrates that RVFE avoids warm-up instability by design.

**Answer to Q2.** RVFE does not rely on MSE pretraining to reach low regret.
**Answer to Q3.** Existing methods are sensitive to warm-up; RVFE is robust.

# G ANNEALING ROBUSTNESS: IS RVFE SENSITIVE TO HYPERPARAMETERS?

We further demonstrate the regret sensitivity of RVFE with respect to its annealing rates $\gamma_\downarrow, \gamma_\uparrow$. These parameters control how the energy-based objective adapts during training. Figure 4 shows the regret landscape over the hyperparameter space.

**Conclusion.** RVFE provides practical robustness, enabling plug-and-play use in real applications.

# H TRAINING DYNAMICS: WHEN DO LOSSES DIVERGE?

To understand how different objectives behave during training, we track both prediction loss and decision loss for six benchmark tasks. Figure 5 shows the loss curves for each.

In the early epochs, prediction loss and decision loss often decrease together, consistent with the warm-up intuition. However, after several iterations, their descent directions diverge. In some cases, reducing prediction error leads to increased regret. This aligns with our theoretical analysis of turning points—locations in the loss landscape where improvements in prediction push the decision across a boundary.

**Answer to Q1.** The loss curves and gradient norms in Fig. 5 show that RVFE avoids the divergence of $\|\nabla_\theta L_{\text{dec}}(\theta)\|$ observed for other decision-focused losses. In MSE and SPO+, the gradient norm

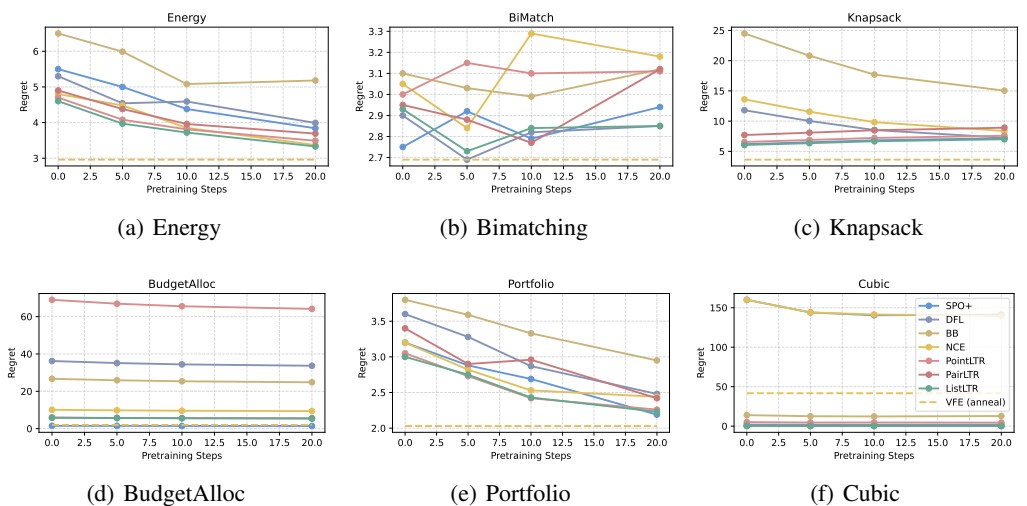

Figure 3: Regret of each method under different pretraining steps. RVFE (orange line) does not require pretraining and consistently outperforms tuned baselines.

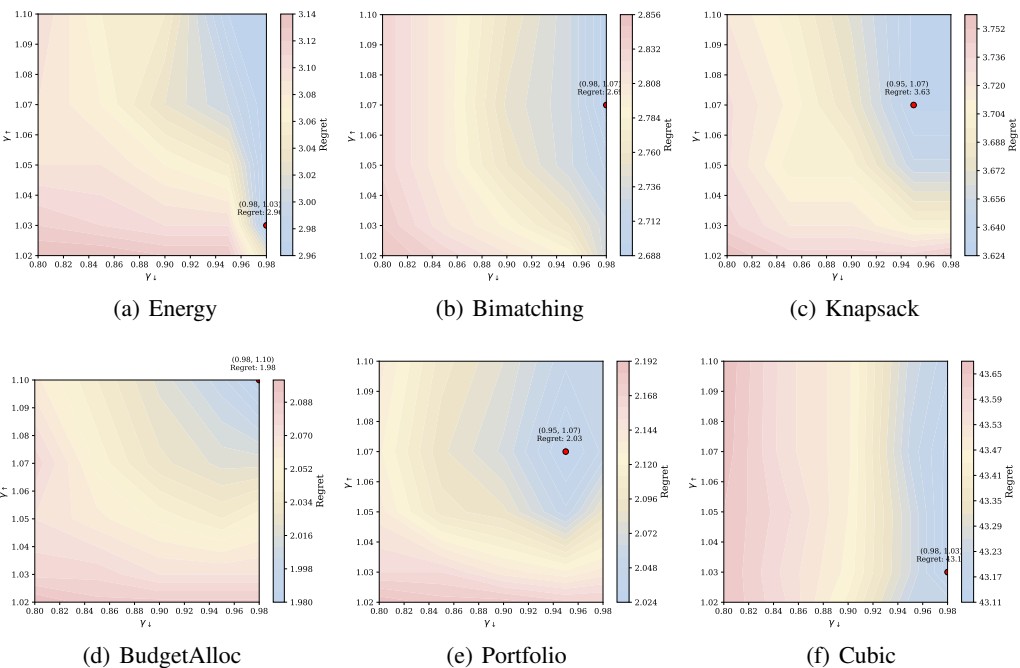

Figure 4: Sensitivity of RVFE to its annealing parameters $\gamma_\downarrow, \gamma_\uparrow$. A wide range of settings yield low regret.

spikes in the early epochs and remains large near the separation point, consistent with the intuition that the decision gradient becomes high-variance and poorly aligned in this region. In contrast, RVFE maintains a moderate gradient norm and a tighter coupling between $L_{\text{pred}}$ and $L_{\text{dec}}$, with both losses decreasing jointly before the separation point and stabilizing afterward. This behavior supports our view of the variational free-energy objective as a Lyapunov-like functional that absorbs the instability of the decision loss and smooths the training dynamics.

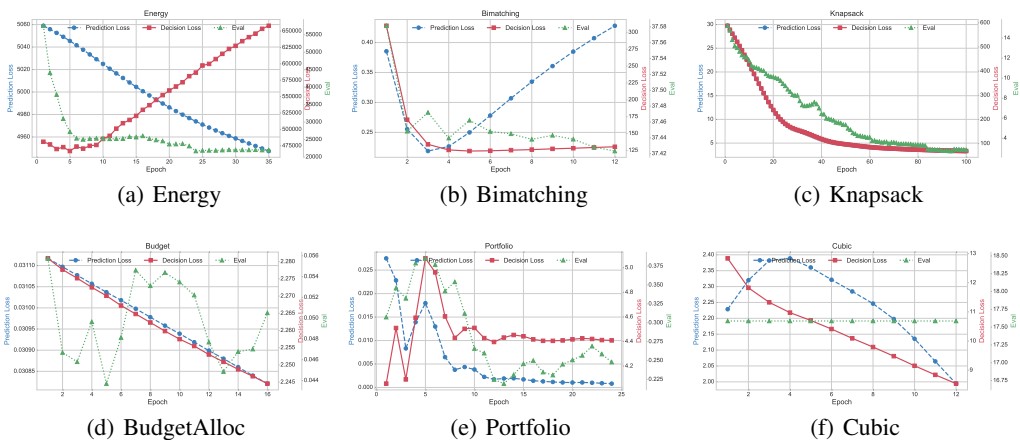

Figure 5: Training curves of prediction loss, decision loss, and final evaluation metric. Disagreement between losses confirms separation point instability.

