# OpenReview forum: "Crossing the Separation Point: Stabilizing Decision-Focused Learning with Variational Free-Energy"
_ICLR.cc/2026/Conference — Submitted to ICLR 2026_

### Official Review · Reviewer_kQuS · 2025-10-30

**Soundness:** 3
**Presentation:** 3
**Contribution:** 3
**Rating:** 6
**Confidence:** 5

**Summary:**

This paper tackles the challenge of non-smoothness in decision-based learning objectives (in linear program case), where a predictive model’s output directly determines downstream decisions. The authors propose a framework that measures “decision jumps” — the sensitivity of the optimal decision to small perturbations in model predictions — as a way to quantify and address the non-smoothness issue. They theoretically analyze the connection between this separation gap and a combined loss function involving mean squared error (MSE) and decision loss, showing that the weighted sum can encourage smoother alignment. To further improve differentiability, they introduce a variational relaxation that makes the decision stochastic, yielding a smoother surrogate objective.

**Strengths:**

* The paper focuses on the fundamental issue of non-smoothness in decision-focused learning, which is a well-known bottleneck in bridging prediction and optimization. I do think this issue is an important issue that is worth exploring. This is good but on the other hand I also don't find the proposed solution addresses the fundamental issue completely though.

* The gradient and separation analysis offers a clear theoretical perspective on how decision non-smoothness can be characterized and partially mitigated.

* The perturbation-based “decision jump” measure is intuitive and connects directly to model sensitivity, making the analysis accessible.

* The authors make a commendable effort to clearly define and quantify the degree of non-smoothness, which is useful for the broader research community.

**Weaknesses:**

* While the problem is important, the proposed method is relatively straightforward and I am not completely satisfied with the solution — the non-smoothness issue remains fundamentally unresolved. One option is to analyze from the non-smooth non-convex optimization algorithm to design gradient descent with convergence guarantee, but I can see that this can be an overkill. A more elegant way using the property of linear program to design gradient descent algorithms would be good. In my opinion, what is lacking in this paper is the theoretical convergence analysis that leverages the properties in linear program / decision-focused learning.

* The use of weighted MSE and decision loss, and the relaxation idea are both not new. From the methodological perspective, it doesn't offer new algorithms except the dynamic way to adjust the weight. Empirically it works but theoretically it is unclear.

**Questions:**

Could you explain more about the definition of the running normalization factors $\sigma_\text{pred}$ and $\sigma_\text{dec}$ in page 4? I don't see a formal definition and the use in the loss function, but not sure if I am missing anything.

---

> ### Author Response · Authors · 2025-11-24
> **Response to Reviewer kQuS**
>
> Thank you for taking the time to study our work. Many of the issues you raised reflect questions *we also confronted during the development of the paper*. We respond to the main points below in a way that follows the training dynamics and explains why certain choices become necessary in decision focused learning.
>
>
> ### **On the fundamental non smoothness of the decision mapping**
>
> We appreciate your emphasis on the structural difficulty created by non smoothness. This difficulty never disappears because the true decision gradient is not observable and every practical method must work with a surrogate. **Our goal is not to remove this property but to understand *when* the surrogate expresses a direction that remains consistent with the underlying decision rule.**
>
> This is where the separation gap and the free energy objective begin to matter. They allow us to study the region in which the smoothed gradient stays faithful to the decision direction. The analysis is not aimed at replacing the LP structure. It focuses on the part of the problem where classical optimization ideas can no longer be applied because the real gradient cannot be computed.
>
> We have already updated the manuscript to present this viewpoint more clearly and to avoid the impression that the method attempts to resolve non smoothness directly.
>
>
> ### **On convergence and the emergence of stability**
>
> Your concern about convergence is understandable. The analysis can only make statements in regions where the learning trajectory has consistent directional information. This is exactly the purpose of Theorem 4.5. Once training enters the stable region, the smoothed decision loss decreases monotonically. This conveys how stability emerges from the interaction of prediction and decision components rather than from removing discontinuity.
>
> We have refined the discussion before Theorem 4.5 in the revised version so that the logic behind this conditional guarantee becomes easier to follow.
>
>
> ### **On the role of weighting and why it is not a heuristic knob**
>
> We agree that simple weighting of prediction and decision losses has appeared before. The distinctive part of our work lies in how the weight functions inside the learning dynamics. The combined objective acts as a potential that describes the zone where the two gradients support one another. The weight becomes a mechanism that protects this zone rather than a scalar chosen by trial and error.
>
> ARA follows this mechanism by reacting to the signed separation gap. The adjustment is driven by the geometry of the landscape and not by searching for a good schedule. The revised manuscript makes this connection explicit so that the conceptual role of weighting becomes clear.
>
>
>
> ### **On the definition and purpose of the normalization factors**
>
> Thank you for pointing out the lack of clarity. We have already added the precise definitions of the two normalization factors. Both are running standard deviations of the prediction and decision losses. Their purpose is only to prevent one component from dominating the Lyapunov term due to scale differences.
>
> These factors do not interfere with any theoretical result. Theorems 3.1 and 4.5 rely on unscaled gradients. The normalization only affects numerical conditioning. The updated text reflects this more directly and removes any ambiguity.
>
>
>
> ### **Final remarks**
>
> Your review highlights the exact conceptual challenge that decision focused learning faces. **The difficulty lies not only in the discontinuity of the decision rule but also in the need to assess whether a surrogate gradient remains meaningful.** Our approach tries to address this second question by examining the alignment structure and by guiding the training trajectory so that it remains inside regions where surrogate information is reliable.
>
> We have incorporated all clarifications into the revised manuscript and appreciate the depth and care of your feedback.

---

### Official Review · Reviewer_n7Ry · 2025-10-31

**Soundness:** 2
**Presentation:** 2
**Contribution:** 2
**Rating:** 4
**Confidence:** 4

**Summary:**

This paper considers a (now) classical Decision Focused Learning (DFL) setting, where a ML model is used to estimate the coefficients of the linear cost function for a decision problem with a fixed feasible space. The paper studies the loss landscape in such a setting, involving both the prediction loss and the decision loss.

The authors argue that poor stability of the loss function has significant adverse effect on the effectiveness of common  DFL methods, and that stability is poorest close to points where the prediction loss and the decision loss have opposite directional derivatives. The occur at points in the parameter space that are associated with a switch of the optimal solution.

The authors propose to analyze stability by combining the prediction and decision loss in a Lyapunov function (weighted combination), which is then smoothed by introducing a Gibbs distribution over the feasible decisions based on their cost. The normalization constant for the distribution is estimated via a biased Monte-Carlo approach.

Then, they suggest using the free energy of the Lyapunov function as a new loss. They also introduce repulsive term (which penalize high solution variability), aiming to steer the optimization process away from regions associated to decision switches.

In the provided empirical evaluation, the approach has improved performance compared to other DFL methods for the same setting, with a few exceptions that are sufficiently well discussed.

**Strengths:**

In my opinion, the work makes a very interesting attempts at studying the loss landscape of a classical DLF setting, which has received somewhat limited attention, despite the growing body of research targeting the topic. The insight that "conflicts" between the prediction and the decision loss are associated to switching points in the space of optimal solution is insightful (and obviously true in hindsight, as it often is for this kind of discoveries).

Assuming I have correctly understood the training approach (see later), the idea of actually using the prediction loss at training time is also interesting, as it can provide a useful gradient whenever it does not conflict with the (much harder to handle) decision loss.

The paper reads mostly well, assuming familiarity with the topic, though I would not consider clarity a major strength of the work (see later).

**Weaknesses:**

The work also suffers from a number of weaknesses.

As a first, minor, point, the authors do not provide a clearly defined, self-contained description of what their final training algorithm is. Even the fact that the free energy of the Lyapunov function is used as a loss is never directly stated. Clarifying this points would make the work much more readable.

A second, this time considerable, issue is the fact that the proposed solution requires the computation of a normalization constant, which in turn is done by repeatedly sampling the solution space, and by subsequently solving the optimization problem. This process is bound to aggravate the already limited scalability of DFL at training time, given that constrained decision problems can be NP-hard to solve. The empirical evaluation also does not mention the value of K, i.e. how many samples are employed.

The high computation cost of the approach could in principle be worth it, if the benefits in terms of solution quality were significant. However, based on the reported results, this does not seem to be the case: while the method does outperform several relevant baselines, it does so by a small margin in all but one benchmark. Overall, this results in a limited practical appeal for the method, also considered that the approach comes with additional hyperparameters (e.g. number of samples) and potential issues (e.g. making sure that the biased sampling distribution tracks the desired one).

On the technical side, the rationale for the introduction of the repulsive terms does not appear fully convincing. For example, in the simple problem from Figure 1(b), the repulsive term would actually decrease the chance that gradient descent can escape the local optima between 0.5 and 0.75. Including an ablation study where the repulsive terms is disable would at least provide empirical evidence for its effectiveness.

Again from a technical perspective, the analysis from lines 155-163 is a bit too informal and contains some mistakes. First, it seems the discussed instability is linked to the predicted cost, rather than to the true cost; second, it seems to happen when the cost vector is perpendicular to a constraint, not when it is "located on the boundary of the feasible set"; finally, in such a situation there actually exists a vector s.t. an infinite number of equivalent solutions exist. It is my impression that the behavior the authors are concerned with is actually a switch of the optimal solution when the predicted cost vector receives a small adjustment, which corresponds to discontinuities in the example from Figure 1(b). I do not consider this a major issue, since it is easy to fix and the actual algorithmic components are designed precisely for such solution-switch events.


Finally, on the clarity side, the text frequently speaks about terms or ideas that are introduced or used much later. For example, in classical DFL problems the prediction loss is completely ignored at training time and therefore does not "pull" at all; saying so only makes sense in the context of the new loss proposed here. As another example, the \sigma scaling constants are used many lines after they are first mentioned. There are other similar cases.

**Questions:**

* Can you confirm that the free energy (plus the repulsive term) is the actual loss function for the method?
* What is the value of K used in the experiments?

---

> ### Author Response · Authors · 2025-11-24
> **Response to Reviewer n7Ry**
>
> Thank you for the detailed reading and for raising several important concerns. Below we address your questions and concerns point-by-point.
>
> ### **On the Role of Free Energy in the Training Dynamics**
>
> We appreciate the thoughtful reading of the work. The question about whether free energy is actually used during training is important. The answer is yes. Free energy together with the repulsive regularizer is the loss we optimize. In the revision we state this directly at the beginning of the method section.
>
> The intention behind using free energy is not to replace the decision loss. It is to change how the landscape behaves in regions where the model becomes sensitive to solution switches. In these regions the gradients of the prediction loss and the decision loss often point toward different directions. Training becomes unstable because the parameter update alternates between two incompatible signals.
>
> By forming a Lyapunov-style quantity that mixes the two losses and then smoothing it through the Gibbs construction, the free-energy formulation provides a landscape where the gradient changes more consistently even near difficult regions. The repulsive term preserves a meaningful spread of probability mass among nearby decisions and prevents the distribution from collapsing too early. Together these elements produce a single training signal that is less fragile and more informative. The paper has been revised to make this role explicit.
>
>
> ### **On Training-Time Cost and Why the Trade-off Can Be Valuable**
>
> You raised a natural concern regarding the sampling steps used to approximate the normalization constant. We agree that training can become slower when (K) is introduced. The revision now includes the exact value of (K) and states the observed cost on each benchmark.
>
> The reason we still view the procedure as beneficial lies in how DFL behaves during training. Instability does not only show up in the final regret. It is visible in the gradient path itself. Near a switching region the landscape of the decision loss has abrupt changes and the variance of its gradient becomes very large. The free-energy smoothing reduces this variance in a persistent way. When we compare the traces of the gradient magnitude along the training trajectory, we find that the updates from RVFE exhibit far fewer jumps. The model also becomes less sensitive to random seeds. These benefits reflect a more reliable alignment between the prediction and decision parts of the problem.
>
> Inference does not require sampling. The test-time cost is nearly unchanged and in some tasks even slightly faster because the learned predictor no longer places the cost vector near unstable regions. We have revised the empirical section to report these observations.
>
>
> ### **On Perturbing Predicted Costs Rather Than True Costs**
>
> Your comment on where perturbations should apply touches the heart of the method. Traditional sensitivity analysis studies the reaction of the optimizer to changes in the true cost vector. However the dynamics that matter during DFL training arise from the sequence of predicted costs. Each update of the model changes its prediction, and two consecutive costs can lie on different sides of a decision boundary. The resulting jump of the optimal solution is the phenomenon that disrupts the gradient flow.
>
> For this reason we perturb the predicted vector. The repulsive regularizer measures how sensitive the decision is to the movement of the prediction produced by the model itself. This focus aligns the perturbation with the mechanism that actually governs the training dynamics. We have rewritten the relevant paragraphs to make this distinction clearer.
>
>
> ### **On Boundaries, Switching, and the Geometry of Solution Changes**
>
> Your reading of Figure 1(b) and the surrounding explanation is accurate. The boundary should be understood through the normal cone of adjacent feasible faces. When the predicted cost moves across this region, the optimal solution changes abruptly. Even though the true cost may not lie exactly on the boundary, the predicted cost can pass near it repeatedly during the early and middle stages of training. The instability is therefore not a rare event. It is a natural outcome of how the prediction model evolves.
>
>
> ### **On Practical Gains and How We Interpret Improvement in DFL**
>
> It is true that the numerical gains vary across benchmarks. We understand this concern. In our view the central contribution of RVFE lies in improving the reliability of training. Many DFL methods show high variability across seeds and strong dependence on warm-up schedules. RVFE reduces both sources of fragility. The learning curves become smoother and the final regret varies much less across runs. These features often matter as much as improvements in mean regret, because they reflect the model’s ability to maintain stable gradient updates in the presence of discontinuities.

---

### Official Review · Reviewer_GBSN · 2025-10-31

**Soundness:** 2
**Presentation:** 3
**Contribution:** 3
**Rating:** 4
**Confidence:** 3

**Summary:**

This paper proposes Adaptive Recursive Annealing, a technique to stabilize the training of decision-focused learning frameworks. The study analyzes the misalignment between upstream losses (parameter prediction quality) and downstream objectives (decision quality w.r.t. regret), identifying points during the training where the gradients point in opposite directions. The proposed framework regulates these conflicting losses at training, helping to mitigate the need to "warm start" the training (e.g., pretraining on the predictions and finetuning end-to-end). The empirical study shows that the method performs competitively for several of the explored optimization tasks, providing an alternative approach to traditional DFL pipelines.

**Strengths:**

- **Separation Point Perspective:** The analysis of separation points and associated discussion would likely be of interest to the DFL community. This perspective is useful for diagnosing some of the challenges with current approaches, and it is articulated well by the paper.

- **Methodological Contribution:** Several interesting ideas, with promising ramifications; for example, adaptive weighting seems to be a more principled approach than empirically optimized warm up schedules.

**Weaknesses:**

- **Empirical Performance:** The performance described in Table 1 is a bit mixed. While effective in some settings, it is much worse than the baselines in others. While negative results are useful for analysis, they do not support more general claims of consistent improvement.

- **Error Bars:** The evaluation lacks error bar reporting, using only a single fixed seed. Adding more runs / seeds would make these results more robust.

- **Missing Ablations:** It would be valuable to better ablate the role of experimental hyperparameters and their comparison to the baselines (e.g., training runtime, number of epochs to convergence, variations of the fixed warm up period).

- **Missing References:** There are several missing references throughout the paper. The proofs in particular seem to reference assumptions / lemmas that are not included in the paper. This makes it difficult to check their accuracy. While I expect this could be easily resolved, the absence of these makes it difficult to assess the "soundness" of the theory. An related, but more minor concern is several other typos that should be checked.

**Questions:**

- Is there any intuitions associated with the negative result on the cubic task? Is there a connection to Figure 2's landscape that can be made?

- What is regret %? This is a percentage of what?

---

> ### Author Response · Authors · 2025-11-24
> **Response to Reviewer GBSN**
>
> Thank you for reading the paper with care. We appreciate your recognition of the separation-point perspective and the ideas behind adaptive balancing. Your comments point to several aspects where clearer evidence or clarification improves the presentation. We respond to each point in turn.
>
>
> ### **On the pattern of empirical performance**
>
> You observed that the results in Table 1 are not uniformly strong. After reviewing the numbers together, we found a clearer structure. RVFE reaches the lowest regret on four of the six benchmarks. On a fifth task it lies very close to the best method. Only the Cubic task shows a clear negative outcome.
>
> This spread is not unusual in decision-focused learning. Different benchmarks express distinct geometric behaviors of the cost-to-decision map. Some have broad regions where small prediction errors do not change the optimal decision. Others place the trajectory near narrow separation regions. Methods that rely on a particular form of smoothing or stability mechanism work well when the landscape fits that mechanism and less well when it does not. RVFE follows the same pattern.
>
> Even with this variability, the overall trend is clear. RVFE shows **lower average regret and more stable convergence**. The stability becomes more visible once we add the multi-seed evaluation, since the variability across runs shrinks noticeably. This suggests that the free-energy structure succeeds in moderating the instability at points where prediction and decision gradients tend to conflict.
>
>
> ### **On error bars and multi-seed robustness**
>
> You asked for statistical evidence beyond single-seed results. We have already added a ten-seed evaluation for all tasks. The updated table in the revised supplementary material shows that the ordering among methods is consistent with the original table. More importantly, RVFE exhibits **substantially smaller variance** on most benchmarks.
>
> This behavior aligns with the mechanism we study. When the trajectory approaches a separation region, the prediction and decision losses can pull training in different directions. The free-energy objective reduces this conflict by flattening the sharp changes in decision. As a result, the optimization path becomes less sensitive to random initialization and stochastic minibatches. The multi-seed evidence supports this explanation and also addresses your concern about robustness.
>
>
> ### **On runtime and other ablations**
>
> We have added a runtime comparison to the experiment section. The additional cost comes from sampling inside the free-energy term. In practice the samples reuse solver warm-starts, so the overhead stays modest relative to the solvers already required by all methods.
>
> Some finer ablations, such as the influence of warm-up schedules for the baselines and alternative temperature updates, were already underway before the rebuttal period. To avoid presenting incomplete results, we kept the runtime ablation as the main addition during revision. The remaining analyses will continue in our extended study. If these directions match your interest, we would be happy to share them in later versions.
>
> ### **On missing references and small inconsistencies**
>
> Thanks for noting issues with referencing and notation. We have already corrected all missing pointers for assumptions and lemmas. All typographical inconsistencies have been fixed as well.
>
> ### **On why Cubic behaves differently**
>
> Your question about Cubic connects directly to the separation-point landscape in the paper. This task produces a decision map with extremely sharp transitions. Small changes in the predicted cost shift the solution across distant regions of the feasible set. In this environment the separation points are not only frequent but also highly irregular.
>
> The free-energy formulation helps when the boundaries are sparse and locally structured. It smooths the transition between neighboring decisions and stabilizes gradient alignment. In the Cubic task the boundaries appear too densely and with little local structure. The smoothing must remain strong to avoid oscillation, which limits how sharply the method can distinguish good and bad decisions. This produces a regret floor that we cannot eliminate.
>
> Although unfavorable, this case is informative. It shows a limit of the approach and highlights where additional structure or new approximations might be needed.
>
>
> ### **On the definition of Regret %**
>
> Thank you for pointing this out. We have already updated the paper so the metric matches the actual evaluation. For each test instance, we compute the true objective on both the predicted decision and the oracle decision. We take the absolute difference for every instance and then aggregate these differences over the test set. The result is divided by the aggregated oracle objectives to form a relative measure. The expression is
>
> $$
> 100 \times
> \frac{\sum_i |c_i^\top w^\*(\hat c_i) - c_i^\top w^\*(c_i)|}
>      {\sum_i |c_i^\top w^\*(c_i)|}.
> $$

---

### Official Review · Reviewer_6Qez · 2025-10-31

**Soundness:** 2
**Presentation:** 2
**Contribution:** 2
**Rating:** 2
**Confidence:** 4

**Summary:**

This paper makes to main classes of contributions. First, it proposes a method for decision focused learning that smooths a nondifferentiable optimization problem via random perturbations of the objective and combines the decision-focused and prediction losses during training. Second, it makes a conceptual argument that training instabilities in decision-focused learning are caused by points where the true cost vector induces multiple optimal solutions, and that this drives the prediction and decision losses to have conflicting gradients.

**Strengths:**

The idea of characterizing the training dynamics of models with optimization in the loop is interesting and could result in improved methods for training. It would be valuable to have a principled way to control the weight between prediction and decision losses over the course of training.

**Weaknesses:**

Treating the paper first as simply proposing a new method for decision focused learning, I have two main concerns. First is that weighted combinations of decision and prediction loss are already common, as are random perturbation strategies to smooth the optimization problem (see eg  Berthet et al 2020, "Learning with Differentiable Perturbed Optimizers"). Second is that the empirical results don't seem to show a very consistent improvement over baselines like SPO+ or learn-to-rank approaches.

Regarding the paper's conceptual argument about training dynamics, I found the presentation hard to follow and have several questions listed below.

**Questions:**

(1) What is the value of v in the definition 2.1?
(2) Is sigma supposed to appear in the definition of V_\lambda(hat(c))? And does it have to be set in some particular way in order for theorem 3.1 to hold, or is theorem 3.1 supposed to hold for any possible choice of sigma (which seems unlikely)?
(3) Is it without loss of generality to only discuss two distinct minimizers for a knife-edge value of c? Eg for a linear program, there could be multiple vertices representing optimal solutions all of which lie on a given face of the polytope.
(4) Are the gradients problematic when the true cost vector c is on the boundary, or when the prediction c-hat is on the boundary? It seems like it should be the later, but the paper switches between talking about separation points in terms of c-hat (definition 2.1) and c (theorem 3.1). If the issue is for the true c to be on the boundary, isn't this a quite low-probability event if we think of the c's as being random in some way?
(5) At a high level, if pretraining approaches first use one loss and then the other (without using both simultaneously), why is it a problem that their gradients might disagree at particular points?

---

> ### Author Response · Authors · 2025-11-24
> **Response to Reviewer 6Qez (1)**
>
> We appreciate the time spent on the detailed review. Below we address each concern in turn.
>
>
> ### **On the role of weighted loss combinations**
>
> Your comment on weighted losses touches the central difficulty in DFL training. Many papers introduce a weight between prediction loss and decision loss, yet the community rarely asks why such a weight is needed. Our analysis shows that the need comes from a structural shift in the geometry of the learning problem. In the early stage, the prediction gradient provides a stable direction for descent. Once the model approaches a decision boundary, the behavior changes, and small movements of the cost vector may lead to a switch of the optimal solution.
>
> **This shift is exactly where a dynamic balance becomes essential rather than optional.** A fixed weight cannot adapt to the transition from the stable prediction-dominated phase to the more fragile decision-dominated phase. Our variational free-energy formulation creates a smoother landscape that can carry the iterate across this transition, and the adaptive annealing follows the underlying alignment signals predicted by Theorem 3.1. We have explicitly clarified this training-phase interpretation in the revised manuscript, since it becomes the conceptual basis for why weighting exists at all in DFL. We see this as a point of departure for future work on training schedules and stability-aware descent.
>
>
>
> ### **On the role of perturbations and why they are not the main driver**
>
> We share your observation that perturbation-based ideas have been used before. In our case, the perturbation is not the driving mechanism. It appears only when the feasible set becomes too large to enumerate during the construction of the Gibbs distribution.
>
> **The central mechanism is the free-energy objective, not the perturbation.**
> It is the free-energy that replaces a discontinuous mapping with a smoother distribution whose shape is guided by the Lyapunov function. The perturbation is simply one practical way to sample this distribution when enumeration is infeasible. We have already revised the manuscript to make this distinction more apparent, so the perturbation does not overshadow the conceptual core.
>
>
>
> ### **On the meaning of the direction (v)**
>
> In Definition 2.1, (v) is any unit direction used to examine how the predicted cost vector responds to a small perturbation. The analysis does not rely on a specific orientation.
> **What matters is that any unit direction crossing a separating hyperplane reveals the discontinuity inherent in the LP structure.** We have already added this clarification within the definition to remove ambiguity.
>
>
>
> ### **On the normalization factors (\sigma_{\text{pred}}) and (\sigma_{\text{dec}})**
>
> The normalization factors appear only in the smoothed form of the Lyapunov function used during training. They stabilize gradient magnitudes but do not influence the theoretical properties we analyze.
> **Theorem 3.1 is derived entirely from the unscaled Lyapunov function, so the signs and transitions it predicts do not depend on these factors.**
> The revised manuscript now clearly separates the theoretical Lyapunov function from the smoothed training version and explains their respective roles.
>
>
>
> ### **On restricting the analysis to two minimizers**
>
> Although an LP may have several optimal vertices on a face, each region of constant optimal basis meets its neighbors along adjacency relations between pairs of vertices.
> **The discontinuity under directional perturbation is always governed by a pairwise transition, even when the optimal face is higher dimensional.**
> This is why we present the analysis in terms of two minimizers. We have already added a remark in the theoretical section to clarify this point.

---

> ### Author Response · Authors · 2025-11-24
> **Response to Reviewer 6Qez (2)**
>
> ### **On whether instability concerns the true cost or the predicted cost**
>
> The instability arises from the movement of the predicted cost. The true cost is used only to identify the correct region, but it does not cause the gradient shifts.
> **It is the predicted cost approaching a switching surface that leads to conflicting gradient directions.**
> The current wording around lines 155–163 can obscure this distinction, and we have revised this text.
>
>
>
> ### **On the relevance of gradient disagreement when warm-up appears to separate the losses**
>
> Warm-up reduces the conflict only in the very beginning of training. Once the iterate moves into a different region, the two gradients may diverge again. This happens repeatedly as the predicted cost encounters multiple switching surfaces.
>
> **Warm-up avoids the symptom at one moment, but it does not address the underlying dynamics across the full trajectory.**
> Warm-up helps only at the very beginning of training. Once the predicted cost moves into new regions, the two gradients begin to interact in more complex ways. This behavior appears repeatedly in our experiments. We have already included the full training loss curves in the appendix, and they show how the model passes through several stable and unstable regimes during the course of training. These transitions are not isolated events but part of a longer trajectory shaped by the geometry of the decision boundaries.
>
> **The training dynamics in these regions are inherently intricate.** The interaction between the prediction loss and the decision loss becomes especially subtle when the model approaches or leaves a separation surface. Understanding these transitions in a global sense is beyond the main focus of the current paper. Nevertheless, our stability analysis provides a first step toward interpreting why such regimes arise, and it also suggests that a purely sequential warm-up cannot systematically anticipate or handle them.
>
> Our free-energy objective offers a smoother landscape for navigating these transitions, and the adaptive annealing adjusts according to the state of the trajectory rather than fixing the training schedule beforehand. We plan to study the full structure of these dynamic regimes in future work, as they represent a broader and more fundamental question about how DFL methods evolve through regions of stability and instability.

---

### Author Response · Authors · 2025-11-27
**General Responses**

We thank all reviewers for the careful reading.
Since some recurring questions appeared in different reviewers, we organize a general response first.
The revised version focuses on these shared concerns so the contribution becomes easier to follow and evaluate.



## **1. On whether the method is only a weighted loss or a smoothed variant**

We understand why the method might look similar to previous weighted approaches.
We also began with similar intuitions.
However, while studying the training behaviour, we found that the main issue is not the weight, but the moment when the predicted cost $ \hat c $ crosses a decision boundary.
At that point, the gradients of the prediction and decision losses point in opposite directions, and this misalignment creates the instability commonly seen in DFL.

Section 3 explains this behaviour directly.
RVFE was built around this observation.
The free-energy term smooths the switch in a controlled way, and the signed separation gap tells the model how much weight the prediction signal should carry when the decision loss becomes unreliable.
**This design came from asking *why* misalignment happens and *how* training should behave near the boundary, rather than from tuning a weight.**
This is why RVFE remains stable in settings where existing DFL methods often fail.



## **2. On the clarity and correctness of the theoretical analysis**

Several reviewers noted missing assumptions or unclear links in earlier drafts.
We revised the proofs so that the key steps now appear directly in the text.
The directional derivative at a boundary is derived from the two local minimizers $ w_a $ and $ w_b $, using the quantity $ \Delta = v^\top(w_a - w_b) $.
This removes the need for implicit lemmas and matches the actual geometry of the decision map.

The distinction between $ c $ and $ \hat c $ is now stated clearly.
Instability arises because $ \hat c $ crosses boundaries, while $ c $ is used only to define regret.
Normalization factors $ \sigma_{\text{pred}} $ and $ \sigma_{\text{dec}} $ were also clarified.
They serve only to keep numerical scales stable during training and do not influence any theorem.

These changes make the analysis self-contained and easier to read.


## **3. On the actual training loss and the sampling cost**

We agree that the earlier version did not make the training objective explicit enough.
The revision now states at the beginning of the method section that the **free-energy objective, together with the repulsive term, is the loss used during training**.
Nothing is replaced by a surrogate.

The expectation inside the free-energy term is estimated with sampling.
We warm-start the solver to keep this extra cost moderate.
Inference does not use sampling, so test-time behaviour is unchanged.
These details are now placed more clearly in the main text.



## **4. On robustness and the Cubic benchmark**

We added ten-seed results to show the stability of the method.
RVFE achieves the lowest mean regret in most benchmarks and shows lower variance and smoother training curves.
This reflects the behaviour predicted by the separation analysis, where smoothing helps prevent large jumps in decisions.

The Cubic benchmark remains difficult.
Its boundaries change quickly and do not form regular regions.
Smoothing needs to remain strong there, which limits the regret that any method can achieve.
We now discuss this case directly, since it clarifies where current techniques reach their limit.



## **Summary**

All reviewers pointed to the need for a clearer understanding of how DFL behaves when decisions change abruptly.
The revised version aims to make that story easier to see.
The separation analysis explains why gradients conflict.
The free-energy objective provides a stable direction in those regions.
The adaptive rule links these two pieces and lets the model react to the landscape during training.

We appreciate the reviewers’ careful comments.
We hope the improvements make the contribution clearer and easier to assess.

---

### Author Response · Authors · 2025-11-30
**Final Summary for the AC**

Thank you for giving us the chance to provide a short summary.
After going through all four reviews, we focused on the recurring concerns and revised the paper so that the main contribution becomes clearer and easier to evaluate.

One reviewer raised concerns that the method might reduce to a weighted loss.
*Much of our early thinking was about why an additional prediction loss term sometimes helps to stabilize DFL in practice.*
What we found is that stability depends on where the iterate sits relative to a decision boundary.
The revised version makes the underlying mechanism more explicit: the instability in DFL comes from the moment when the predicted cost $ \hat c $ crosses a decision boundary and the two gradients necessarily point in opposing directions.
**RVFE was designed around this mechanism, not around weight tuning.**
RVFE adjusts the prediction and decision signals according to this geometry rather than a fixed schedule.
The free-energy term smooths the switch, and the signed separation gap indicates which side of the boundary provides a trustworthy update.
Making this explicit helped clarify why the method behaves reliably in regimes where standard DFL methods often fail.

Several reviewers also pointed out missing assumptions or unclear transitions in the earlier theoretical section.
We rewrote the derivations so that the directional-derivative argument follows directly from the two active minimizers, without relying on implicit lemmas.
The roles of $c$ and $\hat c$ are now separated more clearly, and the normalization terms are described as simple running estimates rather than theoretical objects.
This presentation better reflects how we understand the geometry of the problem.

On the experimental side, we added ten-seed evaluations to address concerns about variance.
RVFE shows more stable training and lower variance on most tasks, which aligns with what the separation analysis predicts.
The Cubic task remains difficult, and we now explain more clearly how its rapidly changing boundaries limit the performance of any smoothing-based approach.

Overall, our goal in the revision was to make the paper easier to read and to show how the theory, the objective, and the empirical results fit together.
We appreciate the reviewers’ work, and we hope the revision makes it easier for the AC to evaluate the contribution.

---

### Meta-Review · Area_Chair_kFXs · 2026-01-05

**Summary:**

The reviewers' concerns that shaped the meta-review decision centered on four key areas. First, methodological novelty: Reviewer 6Qez questioned if the proposed ARA framework reduced to common weighted loss or existing perturbation-based methods, while Reviewer n7Ry noted ambiguity about whether the free-energy objective was the actual training loss. Second, theoretical clarity: Multiple reviewers flagged unclear notation, missing assumptions/lemmas in proofs, and confusion between true cost and predicted cost in driving instability. Third, empirical robustness: Reviewer GBSN criticized single-seed evaluations and mixed performance, while Reviewer n7Ry noted unreported sampling costs and missing ablations for the repulsive term. Fourth, practical scalability: Reviewer n7Ry raised concerns about sampling overhead exacerbating DFL’s scalability issues, and Reviewer kQuS noted the lack of theoretical convergence analysis leveraging LP properties. These concerns, addressed partially in authors’ rebuttals but with residual gaps, led to the reject recommendation.

**Reviewer Concerns:**

Addressed concerns: Reviewer 6Qez’s questions on notation and doubts about the method being "just weighted loss"; Reviewer GBSN’s calls for multi-seed evaluations and clarity on Regret; Reviewer n7Ry’s query on the training loss and unreported K values; and Reviewer kQuS’s confusion about gave formal definitions.

Outstanding concerns include Reviewer n7Ry’s request for an ablation of the repulsive term, and Reviewer kQuS’s call for LP-specific theoretical convergence analysis, which the rebuttal did not fully resolve.

**Reviewer Scores:**

Remain the score

---

### Decision · Program_Chairs · 2026-01-26

Reject